# Unleashing Large-Scale Video Generative Pre-training for Visual Robot Manipulation

**Hongtao Wu**[†], **Ya Jing**[†], **Chilam Cheang**, **Guangzeng Chen**, **Jiafeng Xu**,
**Xinghang Li**, **Minghuan Liu**, **Hang Li**, **Tao Kong**[‡]
ByteDance Research
{wuhongtao.123,kongtao}@bytedance.com

## Abstract

Generative pre-trained models have demonstrated remarkable effectiveness in language and vision domains by learning useful representations. In this paper, we extend the scope of this effectiveness by showing that visual robot manipulation can significantly benefit from large-scale video generative pre-training. We introduce GR-1, a GPT-style model designed for multi-task language-conditioned visual robot manipulation. GR-1 takes as inputs a language instruction, a sequence of observation images, and a sequence of robot states. It predicts robot actions as well as future images in an end-to-end manner. Thanks to a flexible design, GR-1 can be seamlessly finetuned on robot data after pre-trained on a large-scale video dataset. We perform extensive experiments on the challenging CALVIN benchmark and a real robot. On CALVIN benchmark, our method outperforms state-of-the-art baseline methods and improves the success rate from 88.9% to 94.9%. In the setting of zero-shot unseen scene generalization, GR-1 improves the success rate from 53.3% to 85.4%. In real robot experiments, GR-1 also outperforms baseline methods and shows strong potentials in generalization to unseen scenes and objects. We provide inaugural evidence that a unified GPT-style transformer, augmented with large-scale video generative pre-training, exhibits remarkable generalization to multi-task visual robot manipulation. Project page: https://GR1-Manipulation.github.io

## 1 Introduction

Recently, generative pre-trained models have demonstrated impressive performance in both natural language processing (NLP) and computer vision (CV) (Brown et al., 2020; Chen et al., 2020; He et al., 2022; Touvron et al., 2023). They model sequences of inputs in a generative manner and pre-train on large-scale datasets before finetuning on specific tasks. Large-scale pre-training allows these models to learn general patterns from large datasets and thus enables them to easily generalize to related finetuning tasks with inherited generalizability and robustness. Robotics data is also generative in a sense that the observation is only revealed after the action is taken. However, unlike NLP and CV, robot data is scarce, as its collection often requires costly and time-consuming human demonstrations. Moreover, robot data is multi-modal, including images, robot states, actions, and language instructions. To address these challenges, prior research has delved into diverse pre-training methods, aiming to enhance the learning capabilities of robots (Nair et al., 2022; Radosavovic et al., 2022; 2023; Seo et al., 2023; Parisi et al., 2022; Shah et al., 2023; Lin et al., 2023; Kumar et al., 2022; Liu et al., 2022; Yen-Chen et al., 2020).

In this paper, we adapt similar generative pre-training paradigm for tackling the problem of multi-task language-conditioned visual robot manipulation. We argue that video generative pre-training is a closely related task to robot action learning for the reason that a robot trajectory itself contains a video sequence. The ability to forecast future images based on past images and language instructions empowers the robot to anticipate forthcoming events, thereby facilitating the generation of relevant

---

[†]Equal contribution. [‡]Corresponding authors.

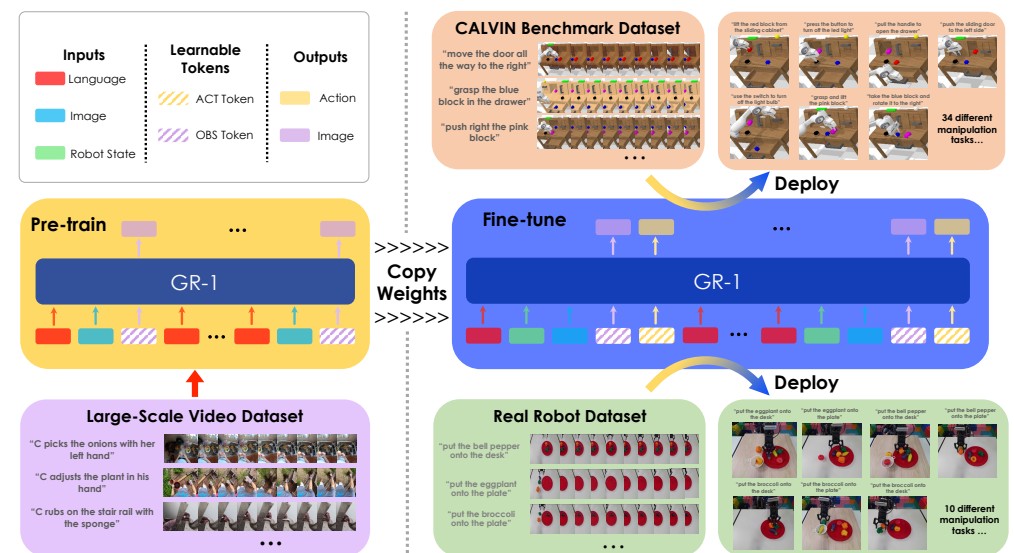

Figure 1: **Overview of GR-1.** GR-1 is first pre-trained on the task of video prediction with a large-scale video dataset. It is then finetuned on robot data to learn multi-task visual robot manipulation.

and suitable actions. To this end, we propose to leverage large-scale video generative pre-training for efficient and effective learning of multi-task visual robot manipulation. We introduce GR-1 (Fig. 1), a GPT-style model which takes as input a language instruction, a sequence of observation images, and a sequence of robot states and predicts robot actions and future images in an end-to-end manner. GR-1 is first pre-trained on video prediction using a large-scale video dataset (Grauman et al., 2022), and subsequently undergoes seamless fine-tuning with robot data.

We perform extensive experiments on the challenging CALVIN benchmark, which contains 34 different manipulation tasks with language instructions. Results show that our method outperforms state-of-the-art baseline methods and improves 1) the success rate from 88.9% to 94.9% and 2) the average length (average number of completed tasks in a row of 5) from 3.06 to 4.21. More importantly, when trained on 10% data of the full dataset, GR-1 achieves a success rate of 77.8% while that of the best baseline method is 66.8%. In the setting of zero-shot unseen scene generalization, GR-1 improves the success rate from 53.3% to 85.4%. When evaluated on unseen language instructions, GR-1 also outperforms all the baseline methods. We perform real robot experiments of end-to-end object transportation and articulated object manipulation to verify the performance of GR-1 in the real world. GR-1 outperforms the comparing state-of-the-art baseline methods and shows promising potentials in out-of-distribution settings, including generalization to unseen scenes and unseen objects. To the best of our knowledge, *GR-1 for the first time shows that a unified GPT-style transformer, augmented with large-scale video generative pre-training, is able to effectively generalize to multi-task visual robot manipulation*. Key contributions of the paper include:

- We show that large-scale video generative pre-training is able to effectively benefit visual robot manipulation learning.
- We present a flexible GPT-style transformer model, GR-1, which allows large-scale video generative pre-training and robot data finetuning with a unified model. Therefore, the model trained on large-scale video datasets can be directly used for robot policy learning.
- We conduct extensive experiments in both simulation and the real world to study the performance of GR-1 in various settings.

## 2 RELATED WORK

### 2.1 LANGUAGE-CONDITIONED VISUAL ROBOT MANIPULATION

Language-conditioned visual robot manipulation is a flexible and intuitive way for non-expert humans to instruct the robot to perform different tasks (Brohan et al., 2022; 2023; Shridhar et al., 2023;

2022; Ahn et al., 2022; Driess et al., 2023; Du et al., 2023; Guhur et al., 2023; Jang et al., 2022; Lynch & Sermanet, 2020; Lynch et al., 2022; Mees et al., 2022a;c; Ha et al., 2023). Some existing methods (Driess et al., 2023; Ahn et al., 2022; Huang et al., 2022) exploit large language models (LLMs) to plan over task domain and pass instructions to low-level action policies to generate robot actions. Ha et al. (2023) uses LLMs to scale up data generation and leverages the diffusion policy (Chi et al., 2023) to learn a language-conditioned policy. Recently, RT-2 (Brohan et al., 2023) proposes to co-fine-tune robot trajectory data and vision-language data together and achieves great generalization capabilities on novel objects and instructions. CLIPort (Shafiullah et al., 2022) and PerAct (Shridhar et al., 2023) learn language-conditioned manipulation by leveraging CLIP (Radford et al., 2021) to connect visual signals and language instructions. Hiveformer (Guhur et al., 2023) proposes to jointly model languages, multiple views, and history with a unified transformer model. These methods predict sparse end-effector keypoint poses and rely on motion planners to plan trajectories, making them less flexible compared to predicting continuous actions. Another line of work learns language-conditioned policies from unstructured play data which contains demonstrations with and without language labels (Lynch & Sermanet, 2020; Mees et al., 2022b;a). These methods leverage sequence-to-sequence conditional variational auto-encoder to generate latent plans which are used to condition the action policy. In contrast, our method uses only demonstrations with language labels and leverages a simple GPT-style transformer to model the trajectory.

## 2.2 SEQUENTIAL DECISION MAKING WITH TRANSFORMERS

There is a growing interest in using transformers (Vaswani et al., 2017) to model sequential decision-making problems. Decision Transformer (DT) (Chen et al., 2021) takes as inputs a sequence of return-to-go, observations, and actions in the past, and outputs actions auto-regressively by leveraging a causal transformer. VIMA (Jiang et al., 2022) enables a multi-modal prompting interface for generalist robot manipulation with an encoder-decoder transformer architecture. GATO (Reed et al., 2022) uses a decoder-only transformer to learn a multi-modal, multi-task, and multi-embodiment generalist policy. RoboCat (Bousmalis et al., 2023) proposes to use heterogeneous robot data to develop a self-improving agent for robot manipulation. It predicts both actions and future images. However, RoboCat does not perform video pre-training and it is goal image-conditioned instead of language-conditioned.

## 2.3 PRE-TRAINING FOR ROBOT LEARNING

Recently, pre-training for robot learning has been an active research topic (Radosavovic et al., 2022; 2023; Nair et al., 2022; Xiao et al., 2022; Liu et al., 2022; Seo et al., 2023; Lin et al., 2023; Kumar et al., 2022; Shah et al., 2023; Jing et al., 2023; Parisi et al., 2022; Escontrela et al., 2023; Laskin et al., 2020; Brohan et al., 2023; Liu et al., 2022; Thomas et al., 2023; Schwarzer et al., 2021; Sun et al., 2023; Yen-Chen et al., 2020; Mendonca et al., 2023; Ma et al., 2022; Bhateja et al., 2023; Karamcheti et al., 2023; Yang et al., 2023). Some methods aim to learn useful visual representations by masked image modeling (Xiao et al., 2022; Radosavovic et al., 2022; Seo et al., 2023; Karamcheti et al., 2023) and contrastive learning (Nair et al., 2022; Sermanet et al., 2018; Jing et al., 2023; Laskin et al., 2020). Another line of work aims to first learn a world model and then train an RL agent with the learned model (Seo et al., 2023; Hafner et al., 2020). SpawnNet (Lin et al., 2023) introduces a two-stream architecture that fuses pre-trained networks with a separate network for learning generalizable policies. VPT (Baker et al., 2022) trains an inverse dynamics model from a small amount of labeled videos and uses it to label large amount of unlabeled videos for action learning in Minecraft. VIPER (Escontrela et al., 2023) leverages pre-trained video prediction models as reward signals for reinforcement learning. The video data in VPT and VIPER are both in-domain data from task environments. In contrast, our method is pre-trained on large-scale non-robotics data which is out-of-domain. Model-based approaches propose to learn a video prediction model and leverage model predictive control (Finn & Levine, 2017; Gupta et al., 2022) or learn an inverse dynamics model (Du et al., 2023) to infer actions. Our method is different from these approaches as it uses a unified model for both video prediction and action prediction. Recently, RPT (Radosavovic et al., 2023) proposes a self-supervised sensorimotor pre-training approach to learn a model of the physical world by predicting masked-out tokens of different modalities. Our method is different in that it adapts video prediction tasks (Yan et al., 2021; 2023) for large-scale pre-training and focuses on language-conditioned multi-tasking.

# 3 METHOD

## 3.1 PROBLEM FORMULATION

**Video Generative Pre-Training.** We leverage language-conditioned video prediction as the video generative pre-training task. Specifically, we pre-train a model $\pi_{\text{pt}}$ to predict the video frame at timestep $t + \Delta t$ given the language description $l$ of the video and a sequence of video frames $\mathbf{o}_{t-h:t}$ from timestep $t - h$ to $t$:

$$\pi_{\text{pt}}(l, \mathbf{o}_{t-h:t}) \rightarrow \mathbf{o}_{t+\Delta t} \tag{1}$$

We assume access to a dataset containing pairs of a video and a language description of the video:

$$v = \{l, \mathbf{o}_1, \mathbf{o}_2, ..., \mathbf{o}_T\}$$

**Multi-Task Visual Robot Manipulation Finetuning.** We formulate multi-task language-conditioned visual robot manipulation as learning a model $\pi_{\text{ft}}$ that maps a language instruction $l$ and a sequence of observation images $\mathbf{o}_{t-h:t}$ and states $\mathbf{s}_{t-h:t}$ from timestep $t - h$ to the current timestep $t$ to an action $\mathbf{a}_t$. In addition, we add future image prediction similar to the video prediction in the pre-training phase:

$$\pi_{\text{ft}}(l, \mathbf{o}_{t-h:t}, \mathbf{s}_{t-h:t}) \rightarrow \mathbf{o}_{t+\Delta t}, \mathbf{a}_t \tag{2}$$

The language instruction $l$ describes the task that the robot is instructed to accomplish, *e.g.* "slide left the red block". The observation sequence $\mathbf{o}_{t-h:t}$ contains visual observation images from the environment. The state sequence $\mathbf{s}_{t-h:t}$ denotes the robot states, *i.e.* the end-effector pose and the binary gripper status. We also assume access to a dataset containing $N$ expert trajectories of $M$ different tasks $D = \{\tau_i\}_{i=1}^N$. Each trajectory consists of a language instruction and a sequence of observation images, robot states, and actions:

$$\tau = \{l, \mathbf{o}_1, \mathbf{s}_1, \mathbf{a}_1, \mathbf{o}_2, \mathbf{s}_2, \mathbf{a}_2, ..., \mathbf{o}_T, \mathbf{s}_T, \mathbf{a}_T\}$$

As will be shown in Sec. 3.3, the model pre-trained by video prediction can be directly extended to finetuning on robot data for learning visual robot manipulation.

## 3.2 ARCHITECTURE

GR-1 (Fig. 1) is a simple GPT-style transformer (Radford et al., 2018) which is able to take different modalities as inputs and outputs future images and actions.

### 3.2.1 INPUT

- **Language input.** The language $l$ is encoded via a text encoder (Fig. 2(a)). Following (Shridhar et al., 2022; 2023), we employ CLIP (Radford et al., 2021) as the language encoder.

- **Visual input.** Visual observations $\mathbf{o}$ are encoded via a Vision Transformer (ViT) which has been pre-trained with MAE (He et al., 2022) (Fig. 2(c)). The output `CLS` token $z_{\mathbf{o}}^{CLS}$ is used as a global representation of the image. The output patch tokens $z_{\mathbf{o}}^{p_{1:i}}$ are used as local representations which are further processed with a perceiver resampler (Jaegle et al., 2021) to reduce the token number.

- **Robot state input.** The robot state $\mathbf{s}$ contains the 6D pose of the robot end-effector $\mathbf{s}_{\text{arm}} \in SE(3)$ and a binary status of the gripper $\mathbf{s}_{\text{gripper}} \in \{0, 1\}$. We use linear layers to encode them (Fig. 2(b)).

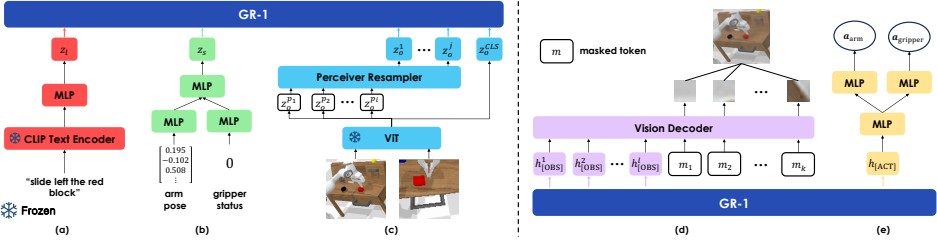

Figure 2: **Encoders and Decoders.** (a) Language encoder. (b) Robot state encoder. (c) Image encoder. (d) Image decoder. (e) Action decoder.

Before being fed into the causal transformer, the embeddings of all modalities are passed through linear layers to align the dimension. For action prediction, we learn an action prediction token embedding to predict the arm and gripper actions. For brevity, we refer to it as `[ACT]`. For video prediction, we learn several observation prediction token embeddings to predict future frames. For brevity, we refer to them as `[OBS]`. During video generative pre-training, the tokens are sequenced as:

$$(l, \mathbf{o}_{t-h}, \texttt{[OBS]}, l, \mathbf{o}_{t-h+1}, \texttt{[OBS]}, ..., l, \mathbf{o}_t, \texttt{[OBS]})$$

When finetuning on robot data, the tokens are sequenced as:

$$(l, \mathbf{s}_{t-h}, \mathbf{o}_{t-h}, \texttt{[OBS]}, \texttt{[ACT]}, l, \mathbf{s}_{t-h+1}, ..., l, \mathbf{s}_t, \mathbf{o}_t, \texttt{[OBS]}, \texttt{[ACT]})$$

Note that the language token is repeated in each timestep to avoid being overwhelmed by other modalities. To inject temporal information, we add learned relative timestep embeddings to the tokens. All modalities at a timestep share the same timestep embedding.

### 3.2.2 NETWORK

We follow the causal attention mechanism typically used in generative pre-trained models, except that all the `[ACT]` and `[OBS]` tokens are masked (Fig. 1). That is, during pre-training, a token can attend to all the tokens in the previous positions except the `[OBS]` tokens; during finetuning, a token can attend to all the tokens in the previous positions except the `[ACT]` and `[OBS]` tokens. We refer the reader to Radford et al. (2018) for more details about causal transformers.

### 3.2.3 OUTPUT

For video prediction, we attach a transformer decoder consisting of self-attention blocks and multi-layer perceptrons (MLPs). The decoder operates on the outputs corresponding the `[OBS]` tokens and mask tokens (Fig. 2(d)). Each mask token is a shared and learnable embedding added with a corresponding positional encoding. The output corresponding to a mask token reconstructs a patch of the predicted future image. Following He et al. (2022), the loss function $L_{\text{video}}$ computes the mean squared error (MSE) between the reconstructed and original images in the pixel space. The outputs from the `[ACT]` tokens are passed through linear layers to predict the arm and gripper actions (Fig. 2(e)). Since the arm action is continuous, we use Smooth-L1 loss $L_{\text{arm}}$ for training. Gripper actions are optimized using Binary Cross Entropy (BCE) loss $L_{\text{gripper}}$.

### 3.3 TRAINING

We first pre-train GR-1 on video prediction and then finetune it on robot data (Fig. 1). In both pre-training and finetuning phases, we freeze the CLIP text encoder and MAE image encoder.

**Pre-Training.** The data for the large-scale video generative pre-training are sourced from the recently proposed Ego4D dataset (Grauman et al., 2022) which contains massive-scale human-object interactions. Ego4D contains more than 3,500 hours of data. Each video clip also contains a natural language annotation describing the behavior of the person in the video. We crop a short clip with a duration of 3s from each video. With this strategy, a total of 800,000 video clips, containing 8M frames, are collected. During pre-training, we randomly sample video sequences and train GR-1 to predict $\mathbf{o}_{t+\Delta t}$ (see the left part of Fig. 1). The network is optimized with causal video prediction loss $L_{\text{video}}$.

**Robot Data Finetuning.** The pre-trained model is finetuned by randomly sampling sequences from the robot dataset and optimizing GR-1 end-to-end with causal behavior cloning loss *and* video prediction loss:

$$L_{\text{finetune}} = L_{\text{arm}} + L_{\text{gripper}} + L_{\text{video}} \tag{3}$$

## 4 EXPERIMENT

We perform experiments on the challenging CALVIN benchmark (Mees et al., 2022c) and a real robot. We aim to answer three questions: 1) Is GR-1 effective on visual robot manipulation? 2) Does GR-1 work on real robots? 3) Can GR-1 handle challenging settings including small dataset,

Table 1: CALVIN Benchmark Results.

| Method | Experiment | Tasks completed in a row | | | | | |
|---|---|---|---|---|---|---|---|
| | | 1 | 2 | 3 | 4 | 5 | Avg. Len. |
| MCIL | ABCD→D | 0.373 | 0.027 | 0.002 | 0.000 | 0.000 | 0.40 |
| RT-1 | ABCD→D | 0.844 | 0.617 | 0.438 | 0.323 | 0.227 | 2.45 |
| HULC | ABCD→D | 0.889 | 0.733 | 0.587 | 0.475 | 0.383 | 3.06 |
| MT-R3M | ABCD→D | 0.752 | 0.527 | 0.375 | 0.258 | 0.163 | 2.08 |
| GR-1 (Ours) | ABCD→D | **0.949** | **0.896** | **0.844** | **0.789** | **0.731** | **4.21** |
| MCIL | ABC→D | 0.304 | 0.013 | 0.002 | 0.000 | 0.000 | 0.31 |
| RT-1 | ABC→D | 0.533 | 0.222 | 0.094 | 0.038 | 0.013 | 0.90 |
| HULC | ABC→D | 0.418 | 0.165 | 0.057 | 0.019 | 0.011 | 0.67 |
| MT-R3M | ABC→D | 0.529 | 0.234 | 0.105 | 0.043 | 0.018 | 0.93 |
| GR-1 (Ours) | ABC→D | **0.854** | **0.712** | **0.596** | **0.497** | **0.401** | **3.06** |
| RT-1 | 10% data | 0.249 | 0.069 | 0.015 | 0.006 | 0.000 | 0.34 |
| HULC | 10% data | 0.668 | 0.295 | 0.103 | 0.032 | 0.013 | 1.11 |
| MT-R3M | 10% data | 0.408 | 0.146 | 0.043 | 0.014 | 0.002 | 0.61 |
| GR-1 (Ours) | 10% data | **0.778** | **0.533** | **0.332** | **0.218** | **0.139** | **2.00** |
| RT-1 | unseen lang | 0.494 | 0.222 | 0.086 | 0.036 | 0.017 | 0.86 |
| HULC | unseen lang | 0.715 | 0.470 | 0.308 | 0.199 | 0.130 | 1.82 |
| MT-R3M | unseen lang | 0.512 | 0.249 | 0.106 | 0.040 | 0.017 | 0.92 |
| GR-1 (Ours) | unseen lang | **0.764** | **0.555** | **0.381** | **0.270** | **0.196** | **2.17** |

"push the sliding door to the right side" "lift the red block from the sliding cabinet" "press the button to turn off the led light" "pull the handle to open the drawer" "grasp and lift the pink block" "take the blue block and rotate it to the right"

Figure 3: **CALVIN Benchmark Results.** We show examples of multi-task learning trained on ABCD→D split.

generalization to unseen scenes, generalization to unseen objects, and generalization to unseen languages? We also perform ablation studies to understand how different modules of GR-1 help visual robot manipulation learning. Ablation studies and more results can be found in the appendix. Videos are available on the project page: `https://GR1-Manipulation.github.io`

### 4.1 CALVIN BENCHMARK EXPERIMENT

CALVIN is a challenging benchmark focusing on learning language-conditioned policy for long-horizon robot manipulation (Fig. 3). It contains 34 tasks and features unconstrained language instructions. The environment contains a Franka Emika Panda robot with a parallel-jaw gripper and a desk with a sliding door, a drawer that can be opened or closed, blocks with different colors, an LED and a light bulb that can be turned on or off.

**Experiment Setup.** For action prediction, similar to Mees et al. (2022c), we train GR-1 to predict delta XYZ positions and delta Euler angles for arm actions and binary gripper actions. The training dataset contains over 20k expert trajectories paired with language instruction labels. Note that the CALVIN dataset contains 24 hours of teleoperated undirected play data. To simulate a real-world scenario, only 1% of the data contains crowd-sourced language instruction labels, on which we train our method. We perform experiments on two splits of data: ABCD→D and ABC→D. A, B, C, and D stand for four different environments (Fig. 4). The four environments are different in desk colors and object configurations. In ABCD→D, we train models with data from all four environments and evaluate in environment D. In ABC→D, models are trained with data from environments A, B, and C and evaluated in environment D which is unseen during training.

**Baseline Methods.** We compare with four baseline methods: MCIL (Lynch & Sermanet, 2020), RT-1 (Brohan et al., 2022), HULC (Mees et al., 2022b), and a multi-task version of R3M (Nair et al., 2022). RT-1 (Brohan et al., 2022) is a state-of-the-art method that uses convolution layers and transformers to generate actions in an end-to-end manner. It uses FiLM layers to condition the convolution layers with the pre-trained embedding of the language instruction. MCIL and HULC take a hierarchical approach which first generates latent plans and conditions the plans on the policy to predict actions. These two methods take as inputs language instructions and observation images taken from the static and gripper cameras. To showcase the effectiveness of video generative pre-training, we compare with another pre-training method R3M (Nair et al., 2022) which is also pre-trained on Ego4D dataset. We use R3M to encode the observation images and leverage a GPT-style transformer to output actions. We freeze the R3M image encoder during training as in Nair et al. (2022). We denote this multi-tasking method as MT-R3M. MCIL and HULC are trained on the full CALVIN dataset containing data with *and* without language annotations. RT-1, MT-R3M, and our method are trained on the data with language annotations which accounts for 1% of the full dataset. We refer the reader to Mees et al. (2022c) for more details on the dataset and language annotations.

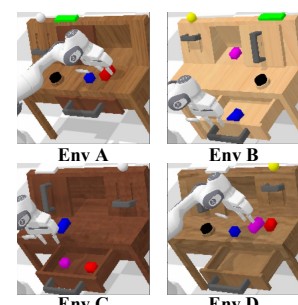

Figure 4: The positions of the sliding door, LED, bulb, light switch, and button are different across the four environments.

**Multi-Task Learning.** We first perform experiments on the ABCD→D split. Quantitative results are shown in Tab. 1. Qualitative results are shown in Fig. 3 and 8. GR-1 outperforms all the baseline methods on sequentially completing 1, 2, 3, 4, and 5 tasks in a row. The average length in the last column, computed by averaging the number of completed tasks in a row of 5 in all the evaluated sequences, shows the long-horizon capability in a comprehensive way. GR-1 outperforms all the comparing baseline methods and improves the best baseline method from 3.06 to 4.21. This demonstrates 1) the superiority of our method on long-horizon multi-tasking and 2) the effectiveness of video generative pre-training.

**Zero-Shot Unseen Scene Generalization.** We also perform experiments on the ABC→D split to evaluate the capability of zero-shot unseen scene generalization. Results are shown in Tab. 1. GR-1 substantially improves the performance in terms of success rate and average length. Specifically, GR-1 achieves a success rate of 85.4%, while that of the best baseline method is 53.3%. This demonstrates that GR-1 possesses strong zero-shot unseen scene generalization capability. We hypothesize that the zero-shot capability of our method is owing to pre-training on large-scale egocentric video clips with rich human-object interactions. This provides consistent and robust visual-textual alignment across different environments for generalization.

**Data Efficiency.** Robot data is expensive and scarce compared to vision-language data. To study the data efficiency, we train on 10% data of the full training dataset from ABCD→D split. Specifically, we sample 66 trajectories for each of the 34 tasks, *i.e.* 2244 trajectories, from the total 22,966 training trajectories. Results are shown in Tab. 1. The performance of all methods degrades compared to training on the full dataset. The best baseline method, *i.e.* HULC, achieves a success rate of 66.8% and an average length of 1.11. GR-1 significantly outperforms all the comparing baseline methods, achieving a success rate of 77.8% and an average length of 2.00. This highlights that GR-1 is efficient when it comes to data. And this is very important as it allows GR-1 to quickly learn skills without collecting a large amount of data.

**Zero-Shot Unseen Language Generalization.** To investigate whether GR-1 can generalize to unseen language instructions, we use GPT-4 (OpenAI, 2023) to generate 50 synonymous instructions for each of the 34 tasks and randomly sampled from them during evaluation (Li et al., 2023). See Tab. 6 for some examples. For GR-1 and all the comparing baseline methods, we use the model trained on ABCD→D split for evaluation and test on the same sampled set of unseen language instructions. The results are shown in Tab. 1. The performance of all the methods drops when evaluated on unseen language instructions. GR-1 outperforms all the comparing baseline methods. We hypothesize that this generalization capability attributes to 1) being exposed to diverse languages in the large video dataset during pre-training and 2) freezing the strong CLIP text encoder during the whole training process to retain its powerful text encoding capability.

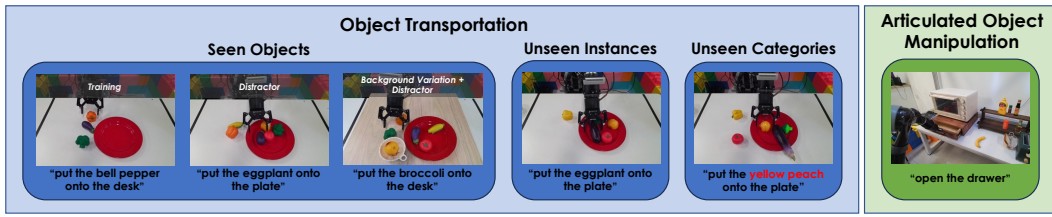

Figure 5: **Real Robot Experiments.** We perform real robot experiments of object transportation and articulated object manipulation.

Table 2: Real Robot Experiment Results.

| Method | Object Transportation | | | Articulated Object Manipulation |
|---|---|---|---|---|
| | Seen Objects | Unseen Instances | Unseen Categories | |
| RT-1 | 0.27 | 0.13 | 0.00 | 0.35 |
| MT-R3M | 0.15 | 0.13 | 0.10 | 0.30 |
| GR-1 (Ours) | **0.79** | **0.73** | **0.30** | **0.75** |

## 4.2 REAL ROBOT EXPERIMENT

To evaluate how GR-1 works in the real world, we perform real robot experiments of object transportation and articulated object manipulation (Fig. 5). More details and results can be found in the appendix.

### 4.2.1 OBJECT TRANSPORTATION

**Experiment Setup.** In this experiment, the basic scene for training contains a plate and three objects: an eggplant, a broccoli, and a bell pepper (leftmost figure in Fig. 5). We collected 1775 demonstrations of transporting one of the three objects from the plate to the desk or *vice versa*, with the HTC Vive VR system. We evaluate in three settings. In the first setting, denoted as *Seen Objects*, the robot is instructed to transport the three objects appeared in the training data. Besides evaluating in a scene that only contains these three objects as in the training data, we also evaluate in two more unseen scenes with disturbance. In the first disturbed scene, we add a tomato, a corn, and a yellow peach as distractors; in the second one, we further change the background by adding a wooden board and a bowl. These two scenes help us evaluate the robustness of GR-1 against disturbance. In the second setting, denoted as *Unseen Instances*, we evaluate the capability of generalization to unseen object instances. The robot is instructed to transport a novel set of eggplant, broccoli and bell pepper which are unseen in the robot training data. In the last setting, denoted as *Unseen Categories*, we evaluate the capability of generalization to unseen categories. That is, the categories of the transported objects are unseen in the robot data. The robot is instructed to transport a tomato and a yellow peach. Example instructions can be found in Fig. 5 and Fig. 9.

**Results.** Quantitative results are shown in Tab. 2. Qualitative results are shown in Fig. 9. GR-1 outperforms the comparing baseline methods in all the three settings. In all settings, RT-1 and MT-R3M typically fail with picking the wrong object and incorrect placing. Another failure mode of RT-1 is collision with the plate or the desk. GR-1 achieves a high success rate in the setting of seen objects. And its performance only drops modestly in the setting of unseen instances. This showcases that GR-1 possesses powerful generalization to unseen instances. In the most challenging setting of unseen categories, a typical failure mode of GR-1 is that it sometimes mixes up the bell pepper with the peach which has a similar color.

### 4.2.2 ARTICULATED OBJECT MANIPULATION

In this experiment, we aim to evaluate on contact-rich articulated object manipulation. We evaluate GR-1 on manipulation of a drawer (rightmost figure in Fig. 5). We collected 2856 trajectories of opening and closing the drawer for training. Quantitative results are shown in Tab. 2 and qualitative

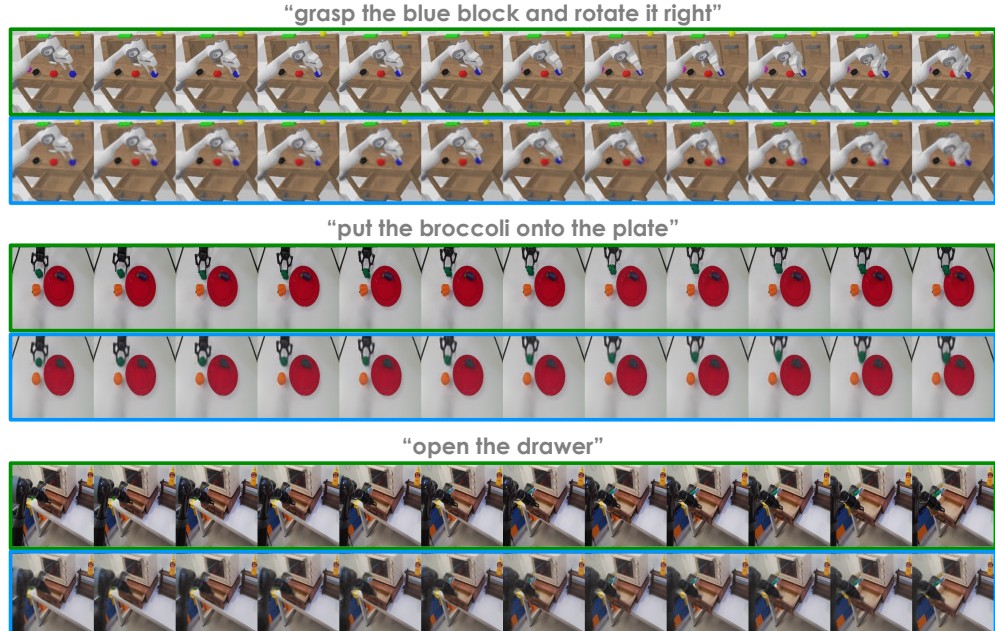

Figure 6: **Video Prediction Results.** The images in green boxes are ground-truth images; the images in blue boxes are predicted images trained by setting $\Delta t = 1$.

results are shown in Fig. 9. GR-1 outperforms the two baseline methods by a large margin. Typical failure modes of GR-1 include 1) failing to completely close the drawer in the closing task and 2) failing to engage with the drawer handle when pulling it out in the opening task.

### 4.3 QUALITATIVE ANALYSIS ON VIDEO PREDICTION

We investigate the video prediction performance of GR-1 finetuned on CALVIN and real robot data. Qualitative results are shown in Fig. 6. And more results can be found in the appendix. GR-1 is able to reconstruct the future image correctly on both CALVIN data and real robot data, although some details (*e.g.* occluded objects) are missing. The video prediction signal can serve as a strong guide for action prediction.

## 5 CONCLUSION

In this paper, we propose to leverage large-scale video generative pre-training for enhancing visual robot manipulation learning. We present GR-1, a GPT-style transformer that takes as inputs a language instruction, a sequence of observation images and robot states, and outputs actions and future images in an end-to-end manner. GR-1 is first pre-trained on language-conditioned video prediction with a large-scale video dataset. Owing to a flexible design, it can then be seamlessly finetuned on robot data to predict actions and future images. Extensive experiments were performed on both CALVIN benchmark and a real robot to verify the performance of GR-1. Results show that GR-1 improves state-of-the-art methods in the settings of multi-task learning, zero-shot unseen scene generalization, small dataset and zero-shot unseen language generalization on CALVIN benchmark. In addition, GR-1 outperforms comparing baseline methods in real robot experiments. By incorporating large-scale video data, we showcase that GR-1 is able to perform robustly in scenes which are disturbed heavily from those in the training data. More importantly, GR-1 is able to generalize to unseen object instances and categories in a zero-shot manner. In the future, we hope to combine video data both with and without languages in training to further enhance the robustness and generalization capability of GR-1. Also, we want to explore the difference of pre-training on videos of any kind v.s. only videos that are more relevant to manipulation. In addition, we plan to scale up the robot data by increasing both the number of robot trajectories in diverse environments and the number of manipulation skills.

**Acknowledgement** We would like to express our gratitude to the authors of Ego4D and CALVIN benchmark. Furthermore, we would like to extend our gratefulness to our colleagues at ByteDance Research for their support throughout this project.

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

# A   APPENDIX

## A.1   NETWORK AND TRAINING DETAILS

The causal transformer of GR-1 contains 12 layers and 12 heads with a hidden size of 384. In total, GR-1 contains 195M parameters, in which 46M of them are trainable. The output layer for action prediction is a three-layer MLP in which the last layer contains two heads for predicting arm actions and gripper actions, respectively. The output layer for video prediction is a transformer consisting of self-attention blocks and linear layers. Following He et al. (2022), the prediction target is normalized patch-wise. Random shift augmentation is applied to the images. In pre-training, we sample equally-spaced frames from video clips in Ego4D dataset to train GR-1. The duration between consecutive frames is 1/3 seconds to ensure that they have sufficient visual difference. At each timestep, GR-1 is trained to predict the image at the next timestep in the sampled frame sequence, $i.e.$ $\Delta t = 1$ in Eq. 1. Robot data is denser compared to the sampled video frame sequences in pre-training. Therefore, when finetuning on robot data, we set $\Delta t = 3$. And we train the network to predict captured images from the static camera and the gripper camera. The input sequence length is 10. We compare on different future steps in Sec. A.4. We apply dropout and use AdamW (Loshchilov & Hutter, 2017) with cosine learning rate decay (Loshchilov & Hutter, 2016) to optimize the network. Hyperparameters for pre-training and finetuning on CALVIN data are shown in Tab 3.

Table 3: Training Hyperparameters

| Hyperparameters | Pre-training | Finetuning |
|---|---|---|
| batch size | 1024 | 512 |
| learning rate | 3.6e-4 | 1e-3 |
| dropout | 0.1 | 0.1 |
| optimizer | AdamW | AdamW |
| learning rate schedule | cosine decay | cosine decay |
| warmup epochs | 5 | 1 |
| training epochs | 50 | 20 |

## A.2   CALVIN BENCHMARK EXPERIMENTS

We follow the evaluation protocol in CALVIN (Mees et al., 2022c) and evaluate 1000 unique sequence instruction chains. For each sequence, the robot aims to *continuously* solve up to 5 tasks by understanding a series of 5 language instructions in a row. If a task is not completed within 360 timesteps, it is considered a failure. The robot receives the next task only if the current one is successfully completed. After each sequence, the robot is set to a neutral position. Fig. 8 shows more rollouts of GR-1 in different settings.

## A.3   REAL ROBOT EXPERIMENTS

In real robot experiments, we use a 7-DoF Kinova Gen2 robot mounted with a RealSense camera on its end-effector. A Kinect Azure camera is used to provide the static view of the scene. In the articulated object manipulation experiment, there are 2 tasks, $i.e.$ opening and closing the drawer. In the object transportation experiment, we evaluate in three settings. In total, there are 10 tasks. They are "put the OBJECT onto the desk" and "put the OBJECT onto the plate". OBJECT={eggplant, bell pepper, broccoli, tomato, yellow peach}. For all the three scenes in the setting of Seen Objects (Fig. 5), the robot is instructed to transport the three seen objects in the training data, $i.e.$ an eggplant, a broccoli, and a bell pepper. The settings of Unseen Instances and Unseen Categories share an identical scene which includes a tomato, a yellow peach, and a novel set of eggplant, broccoli, and bell pepper. The robot is instructed to transport the unseen eggplant, broccoli, and bell pepper in the

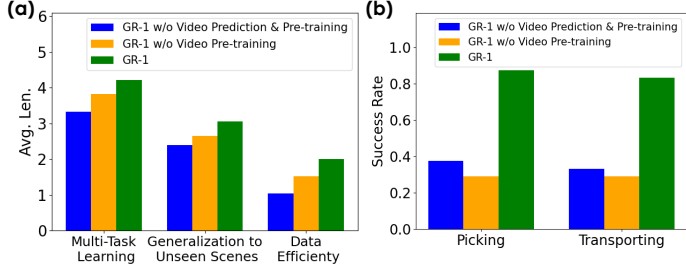

Figure 7: **Ablation Studies.** (a) We show the average length (averaged number of completed tasks in a row of 5) on CALVIN benchmark. (b) We show the success rates of picking and transporting in real robot experiments.

setting of Unseen Instances and the tomato and yellow peach in the setting of Unseen Categories, respectively. In both experiments, we use the same training setting as in the CALVIN experiments except that the batch size and training epochs are changed to 64 and 30, respectively. See Fig. 9 for real robot rollouts.

Table 4: Ablation Studies.

| Pre-Training | Video Prediction | Data | Tasks completed in a row | | | | | |
| --- | --- | --- | --- | --- | --- | --- | --- | --- |
| | | | 1 | 2 | 3 | 4 | 5 | Avg. Len. |
| ✗ | ✗ | ABCD→D | 0.889 | 0.775 | 0.661 | 0.549 | 0.459 | 3.33 |
| ✗ | ✓ | ABCD→D | 0.918 | 0.833 | 0.761 | 0.685 | 0.619 | 3.82 |
| ✓ | ✓ | ABCD→D | **0.949** | **0.896** | **0.844** | **0.789** | **0.731** | **4.21** |
| ✗ | ✗ | ABC→D | 0.823 | 0.609 | 0.425 | 0.318 | 0.225 | 2.40 |
| ✗ | ✓ | ABC→D | 0.815 | 0.651 | 0.498 | 0.392 | 0.297 | 2.65 |
| ✓ | ✓ | ABC→D | **0.854** | **0.712** | **0.596** | **0.497** | **0.401** | **3.06** |
| ✗ | ✗ | 10% data | 0.526 | 0.288 | 0.138 | 0.061 | 0.022 | 1.04 |
| ✗ | ✓ | 10% data | 0.698 | 0.415 | 0.223 | 0.133 | 0.052 | 1.52 |
| ✓ | ✓ | 10% data | **0.778** | **0.533** | **0.332** | **0.218** | **0.139** | **2.00** |

## A.4 ABLATION STUDIES

**Video Prediction & Pre-training.** In this section, we perform ablation studies to study how video generative pre-training helps GR-1 on learning visual robot manipulation. We investigate two variants of GR-1. The first variant, denoted as GR-1 w/o Video Prediction & Pre-training, is trained from scratch and does not predict videos. That is, [OBS] tokens are removed from the input token sequence. The second variant, denoted as GR-1 w/o Video Pre-training, is trained from scratch and retains video prediction. On CALVIN benchmark, we perform experiments on ABCD→D split, ABC→D split, and 10% training data (Fig. 7(a) and Tab. 4). We also perform real-robot experiments in a pick-and-place setting which uses the training scene in Fig. 5. The robot is trained to perform picking an object on the desk/plate and placing it onto the plate/desk. Results are shown in Fig. 7(b). GR-1 outperforms both variants in all experiments. We hypothesize that this is because the large-scale video pre-training helps GR-1 learn an accurate video prediction model which helps the robot understand what shall happen in future steps given the language instruction and previous observations. And this information acts as a strong signpost for the robot to generate pertinent actions for rolling out trajectories. Without pre-training, the video prediction of GR-1 w/o Video Pre-training may not be as robust. On CALVIN benchmark, it outperforms GR-1 w/o Video Prediction & Pre-training but achieves a lower success rate in real robot experiments.

**Different Future Predictions.** We compare the effectiveness of predicting images at different future steps (*i.e.* 1, 3, and 5) on CALVIN benchmark. Results are shown in Tab. 5. Pre-training is not used in this ablation. It is observed that increasing the step from 1 to 3 improves success rates. This may

be because consecutive frames are very similar and predicting frames that are farther away from the current step helps the robot to understand more about the future. But the improvement saturates soon. We hypothesize that this is because the model is trained to predict local actions and predicting frames that are too far into the future may not be able to provide good guidance for immediate local action prediction.

Table 5: Ablation Studies on Video Prediction.

| Future Step | Tasks completed in a row | | | | | Avg. Len. |
|:---:|:---:|:---:|:---:|:---:|:---:|:---:|
| | 1 | 2 | 3 | 4 | 5 | |
| 1 | 0.895 | 0.802 | 0.710 | 0.643 | 0.562 | 3.61 |
| 3 | 0.918 | 0.833 | 0.761 | 0.685 | 0.619 | 3.82 |
| 5 | 0.909 | 0.806 | 0.719 | 0.649 | 0.583 | 3.67 |

## A.5 TASK SUCCESS RATES

To probe into how video generative pre-training helps visual robot manipulation learning, we compare task success rates of GR-1 with GR-1 w/o Video Prediction & Pre-traing. Results are shown in Tab. 7. Tasks with large improvements mostly involve block manipulation. They are difficult because the robot needs to first grasp the correct block and then manipulate it according to the language instruction. With video generative pre-training, the performance of these tasks improves.

We also compare the task success rates with training on 10% data from ABCD→D split to investigate the effect of different data sizes. Similar to the pattern found in the above, when increasing the data size, tasks involving block manipulation show large improvement. Also, the success rate of turning on/off lightbulb also increases substantially.

## A.6 MORE RESULTS

Table 6: Examples of Unseen Language Instructions Generated by GPT-4 (OpenAI, 2023) for the Zero-Shot Unseen Language Generalization Experiment on CALVIN.

| Original | Generated |
|:---:|:---:|
| "use the switch to turn off the light bulb" | "use the switch to stop the light source" |
| "slide the block that it falls into the drawer" | "Move the block ensuring it goes into the drawer" |
| "pull the handle to open the drawer" | "Acquire a grip on the handle to slide the drawer out" |
| "lift the pink block from the sliding cabinet" | "Hoist up the pink block kept in the sliding cabinet" |
| "store the grasped block in the sliding cabinet" | "Hide the item gripped in sliding stash" |
| "take the red block and rotate it to the right" | "Twist the red object to the right" |
| "press the button to turn on the led light" | "press the switch to turn on the glowing LED" |
| "grasp and lift the blue block" | "grasp firmly and raise the blue block" |

Table 7: Task Success Rates (%).

| Task | GR-1 | GR-1 w/o Video Prediction & Pre-training | GR-1 trained on 10% data from ABCD→D split |
|---|---|---|---|
| rotate blue block right | 94.9 | 71.2 | 51.6 |
| move slider right | 99.3 | 99.6 | 86.9 |
| lift red block slider | 98.5 | 91.7 | 50.6 |
| turn off led | 100 | 100 | 95.6 |
| push into drawer | 82.9 | 79.8 | 74.7 |
| lift blue block drawer | 100 | 93.8 | 80.0 |
| lift pink block slider | 97.8 | 93.8 | 56.4 |
| place in slider | 91.3 | 89.1 | 34.8 |
| open drawer | 99.4 | 100 | 94.2 |
| rotate red block right | 98.6 | 81.9 | 45.3 |
| lift red block table | 97.7 | 76.7 | 36.5 |
| lift pink block table | 94.1 | 72.0 | 73.8 |
| move slider left | 99.2 | 99.5 | 90.7 |
| turn on lightbulb | 99.4 | 98.6 | 79.8 |
| rotate blue block left | 97.1 | 71.2 | 66.7 |
| push blue block left | 84.1 | 72.7 | 56.1 |
| close drawer | 99.5 | 98.8 | 91.3 |
| turn off lightbulb | 99.3 | 100 | 79.6 |
| turn on led | 100 | 98.7 | 95.6 |
| stack block | 80.1 | 45.7 | 43.2 |
| push pink block right | 61.8 | 60.3 | 50.0 |
| push red block left | 82.3 | 77.8 | 54.7 |
| lift blue block table | 97.1 | 66.2 | 64.9 |
| place in drawer | 98.9 | 98.6 | 95.0 |
| rotate red block left | 95.3 | 70.5 | 61.5 |
| push pink block left | 89.6 | 84.5 | 73.8 |
| lift blue block slider | 97.0 | 90.1 | 54.7 |
| push red block right | 54.2 | 49.3 | 43.6 |
| lift pink block drawer | 100 | 81.8 | 50.0 |
| rotate pink block right | 91.5 | 79.1 | 51.7 |
| unstack block | 100 | 84.4 | 100.0 |
| rotate pink block left | 96.4 | 70.4 | 66.0 |
| push blue block right | 53.6 | 50.0 | 33.9 |
| lift red block drawer | 100 | 100 | 87.5 |

**Long-Horizon Multi-Tasking: ABCD→D**

"store the grasped block in the sliding cabinet"

"pull the handle to open the drawer"

"grasp and lift the red block"

**Zero-Shot Generalization: ABC →D**

"push the sliding door to the left side"

"use the switch to turn off the light bulb"

"remove the stacked block"

Figure 8: **Rollouts on CALVIN benchmark.**

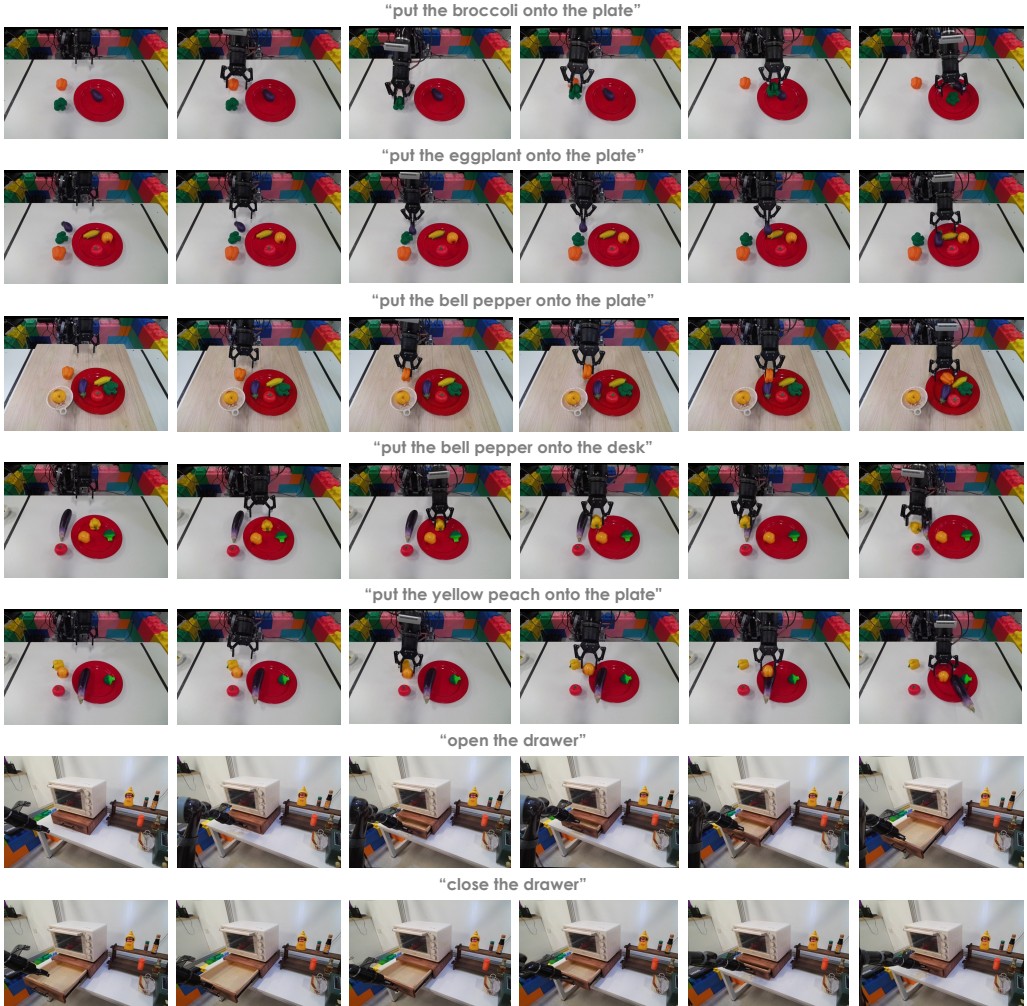

Figure 9: **Real Robot Rollouts.** The first five rows show object transportation rollouts. The last two rows show articulated object manipulation rollouts.

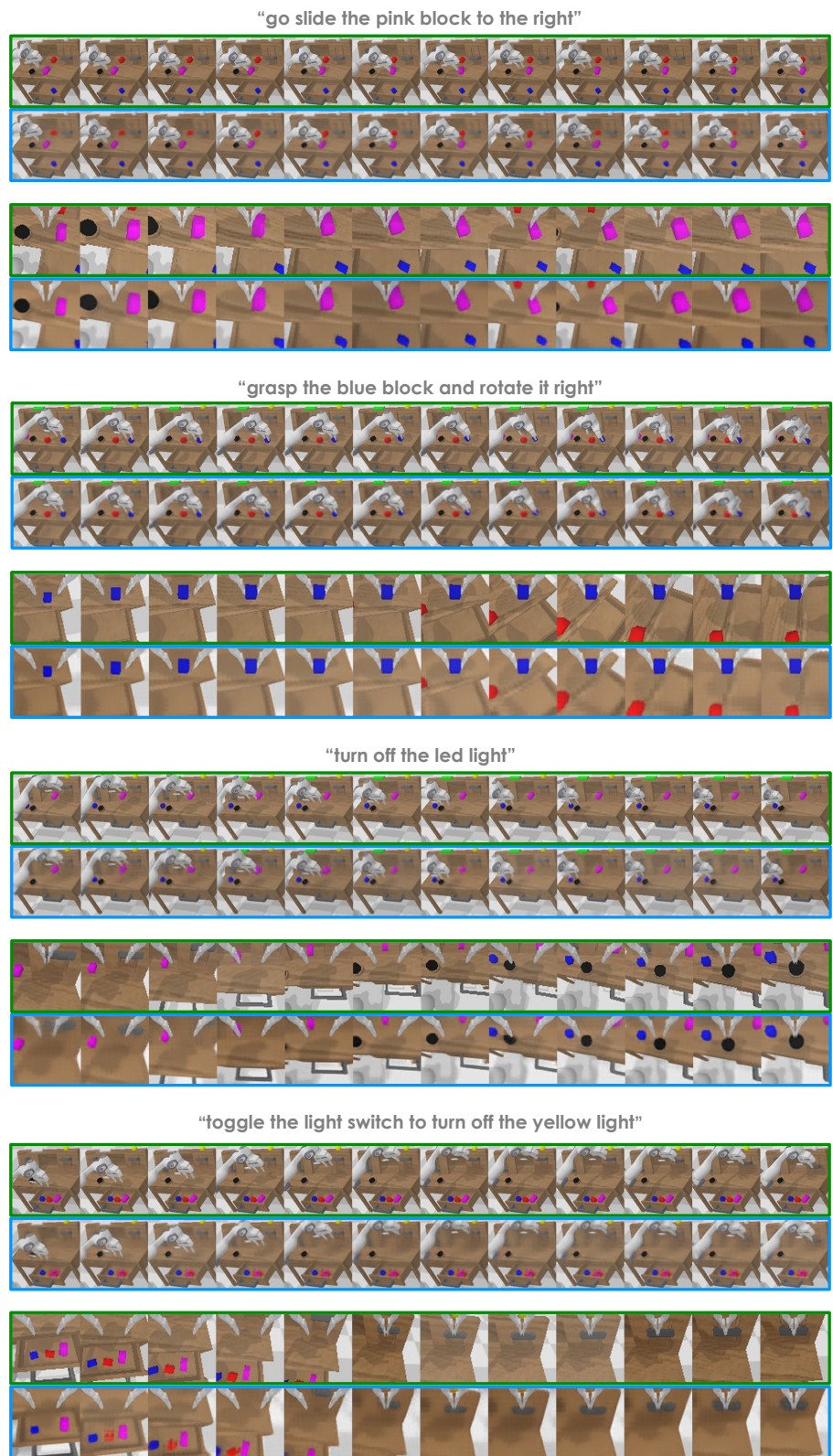

Figure 10: **Video Prediction Results on Calvin Benchmark.** The images in green boxes are ground-truth images; the images in blue boxes are predicted images trained by setting $\Delta t = 1$.

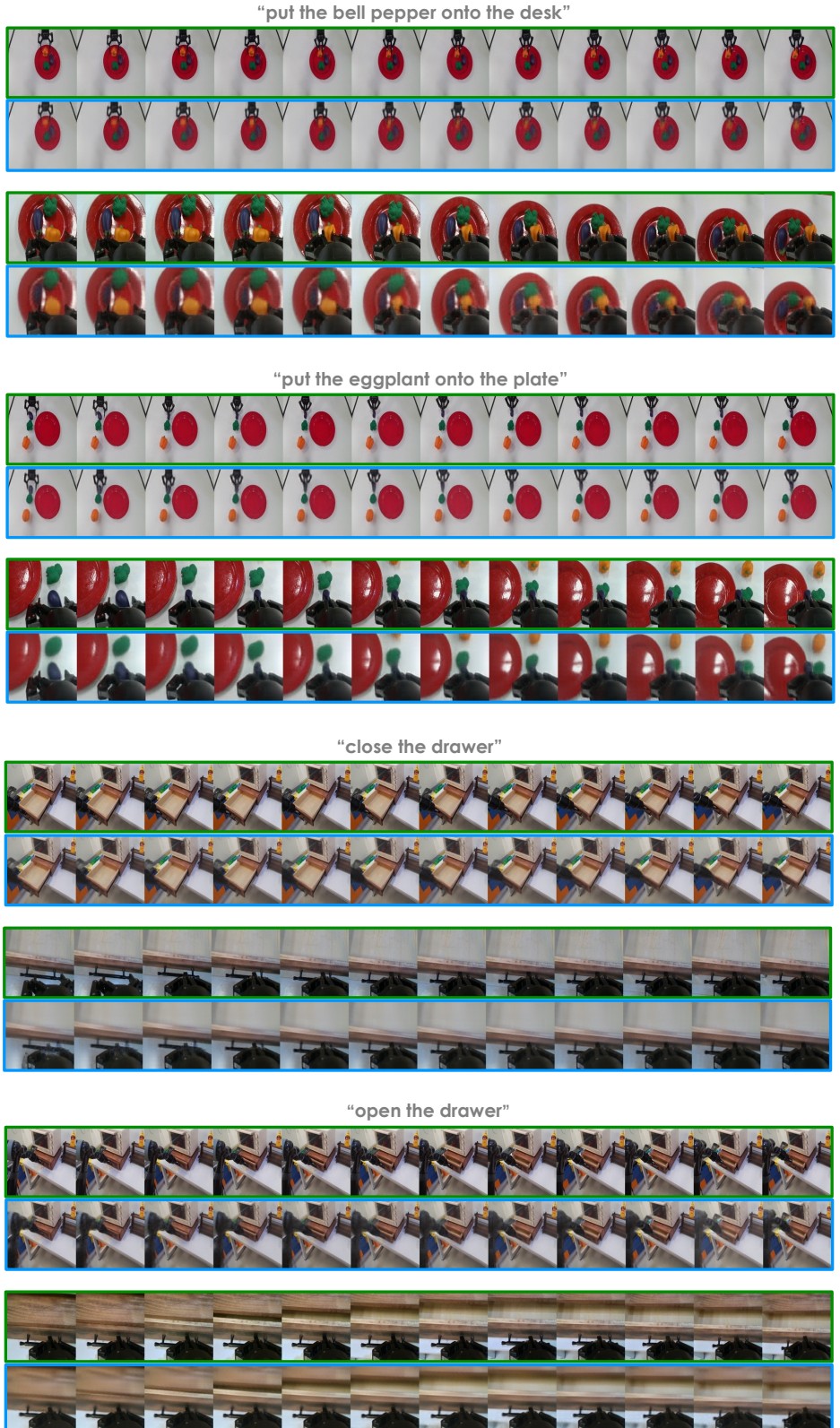

Figure 11: **Video Prediction Results of Object Transportation and Articulated Object Manipulation Experiments in Sec. 4.** The images in green boxes are ground-truth images; the images in blue boxes are predicted images trained by setting $\Delta t = 1$.

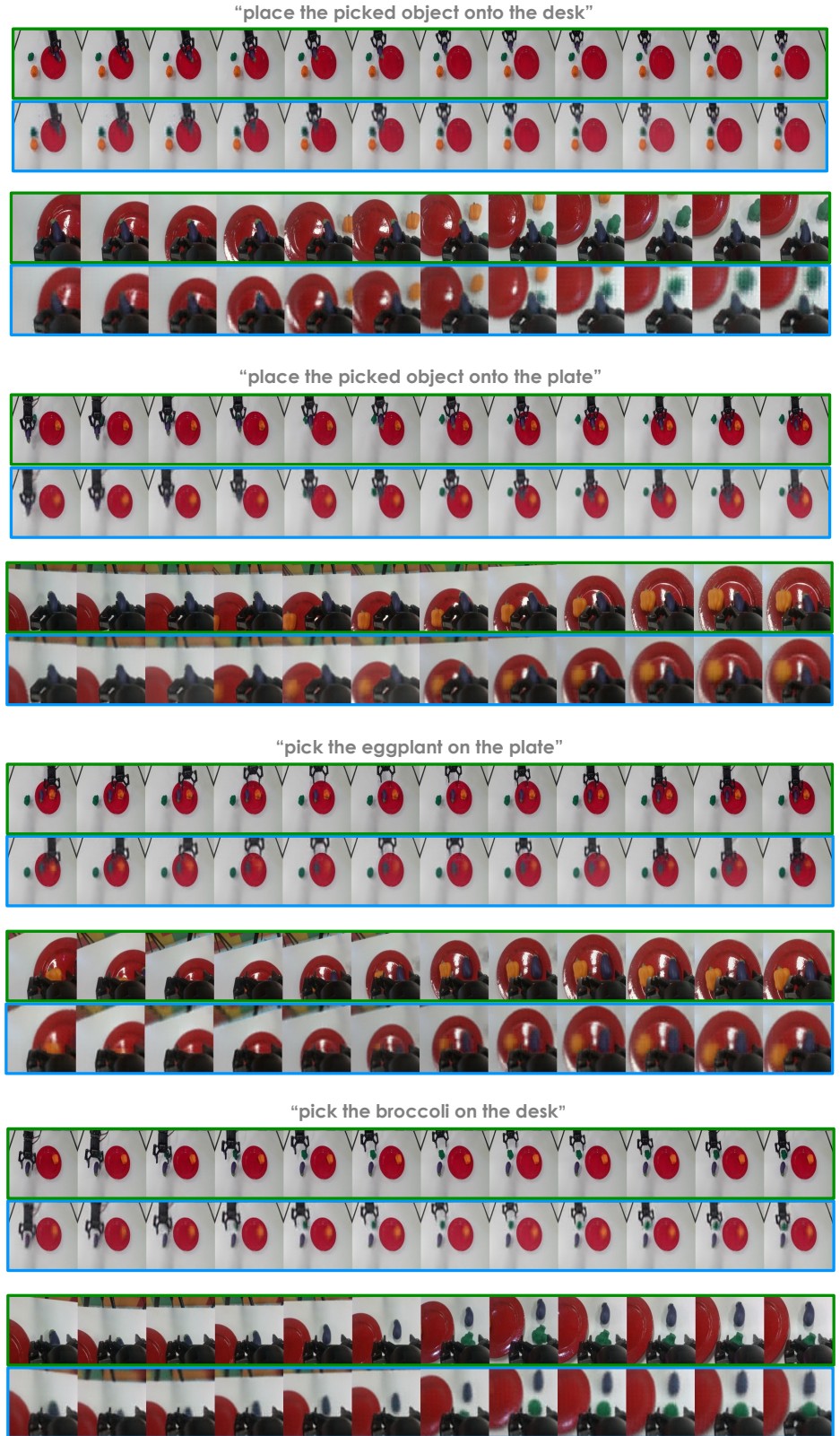

Figure 12: **Video Prediction Results of Pick-and-Place Experiments in Ablation Studies in Sec. A.4.** The images in green boxes are ground-truth images; the images in blue boxes are predicted images trained by setting $\Delta t = 1$.

