# OpenReview forum: "Unleashing Large-Scale Video Generative Pre-training for Visual Robot Manipulation"
_ICLR.cc/2024/Conference — ICLR 2024 poster_

### Official Review · Reviewer_H1qB · 2023-10-30

**Soundness:** 3 good
**Presentation:** 3 good
**Contribution:** 3 good
**Rating:** 8
**Confidence:** 5

**Summary:**

This paper proposes GR-1, a GPT-like autoregressive Transformer model that leverages large-scale video generative pretraining and downstream robotic dataset finetuning to solve language-conditioned robotic manipulation tasks. Specifically, the model is pretrained on Ego4D to reconstruct a future observation from the current observation, and then finetuned on robotic datasets with a combination of video prediction and robotic action prediction losses. Experiments on the Calvin Benchmark and on real robots show that the proposed method achieves better language-conditioned robot manipulation success rates on different task variations.

**Strengths:**

- The paper is clearly presented and easy to read.
- The proposed video generative pretraining approach, combined with downstream finetuning on task-specific robotic manipulation datasets, achieves better language-conditioned robotic manipulation performance than previous baselines like RT-1.
- The effectiveness of the proposed approach not only holds in simulation, but also transfers to real robot tasks.
- The proposed approach shows promises in scaling up to larger-scale video pretraining datasets and more robot manipulation tasks in the future.

**Weaknesses:**

- Is there a plan to open-source the code and checkpoints? Making them available to the community would greatly facilitate efforts to scale up pretraining for robotics.
- Comparing Table 1 and Table 2, the performance on the Calvin Benchmark significantly falls when only using 10% of robotic dataset for finetuning (2k trajectories) compared to using 100% of finetuning dataset (20k trajectories). According to the table, it looks like the primary reason is that the success rate primitive skills (i.e., success rate for "1 task completed in a row") falls from 94.9% to 77.8%. Thus it would be helpful to provide a further analysis into this phenomenon along with the failure modes.
- For zero-shot generalization experiments conducted in the paper, looks like the experiments are primarily conducted on object position variations, color variations, and background variations. It would be helpful to provide an analysis on whether the policy can (1) generalize to longer horizon tasks than the length of the tasks seen during training (e.g., train the policy on 1-5 tasks in a row, test on 6-10 tasks completed in a row); (2) generalize to novel language instructions that are semantically similar to those seen during training but with different wordings (e.g., instead of "lift the red cube", use "pick up the red cube" as instructions; instead of "push the sliding door to the right", use "open the door to the right side" as instructions)
- It would be helpful to provide an ablation on the effect of different quantities of Ego4D pretraining datasets on downstream robot manipulation task performance. Also is there a (possibly smaller) subset of Ego4D dataset for pretraining that is especially helpful for downstream robot manipulation tasks?
- The number of visual features per image from the MAE encoder can be large, and as a result, the current policy might not generalize to very long horizon tasks. It would be helpful to explore using e.g., Perceiver Resampler, to downsample the number of input image features.

For the second to the last weakness points listed above, I wouldn't expect authors to address all of them given the limited rebuttal time, but it would be great if authors could address most of them.





**Update Nov. 23**: Thanks authors for the rebuttal! All of my concerns have been addressed, and I believe that the paper will be very valuable for the vision language & robotics community to scale up generative pretraining efforts for large-scale, open-world robotic manipulation skill learning. The authors also promise to release the code and checkpoints. Thus I've increased my rating and confidence.

**Questions:**

See "weaknesses"

---

> ### Author Response · Authors · 2023-11-23
> **Response to Reviewer H1qB [1/3]**
>
> Dear Reviewer,
>
> Thanks for your constructive comments. Below, we would like to address all the weaknesses and questions in details.
>
> **Q1**: ```Is there a plan to open-source the code and checkpoints? Making them available to the community would greatly facilitate efforts to scale up pretraining for robotics.```
>
> **A1**: Sure! We will release the code and checkpoints. We hope this project can serve as an effective approach to robot pre-training for the community.
>
> **Q2**: ```Comparing Table 1 and Table 2, the performance on the Calvin Benchmark significantly falls when only using 10% of robotic dataset for finetuning (2k trajectories) compared to using 100% of finetuning dataset (20k trajectories). According to the table, it looks like the primary reason is that the success rate primitive skills (i.e., success rate for "1 task completed in a row") falls from 94.9% to 77.8%. Thus it would be helpful to provide a further analysis into this phenomenon along with the failure modes.```
>
> **A2**:
> We visualized the success rate of each task in Tab. 7 of the paper and added discussions in Appendix A.7.
> It can be concluded that the performance on tasks which interacts with blocks (e.g., lift red block slider, place in slider, lift red block table) drops substantially when the data size decreases to 10%.
> These tasks are difficult in a sense that the robot needs to first grasp the correct block and then manipulate it according to the language instruction.
> In addition, the performance on turning on/off the lightbulb also decreases.

---

> ### Author Response · Authors · 2023-11-23
> **Response to Reviewer H1qB [2/3]**
>
> **Q3**: ```For zero-shot generalization experiments conducted in the paper, looks like the experiments are primarily conducted on object position variations, color variations, and background variations. It would be helpful to provide an analysis on whether the policy can (1) generalize to longer horizon tasks than the length of the tasks seen during training (e.g., train the policy on 1-5 tasks in a row, test on 6-10 tasks completed in a row); (2) generalize to novel language instructions that are semantically similar to those seen during training but with different wordings (e.g., instead of "lift the red cube", use "pick up the red cube" as instructions; instead of "push the sliding door to the right", use "open the door to the right side" as instructions)```
>
> **A3**:
> Regarding the task length, we believe there is a misunderstanding here.
> We strictly follow the evaluation protocol in [CALVIN](http://calvin.cs.uni-freiburg.de) [1].
> Each trajectory in the robot training data only contains one single task.
> Therefore, the robot is not trained on "1-5 tasks in a row".
> The evaluation is conducted by instructing the robot to perform up to 5 tasks in a row.
> Thus, the evaluation is already testing the capability of generalizing to longer horizon tasks than the length of the tasks seen during training.
> GR-1 improves the success rate of completing 5 tasks in a row from 38.3% to 73.1%.
> This significant improvement shows its strong long-horizon multi-tasking capability.
>
> Regarding the novel language instructions, we added an experiment to test the generalization capability to unseen languages.
> Specifically, we use GPT-4 to generate 50 synonymous instructions for each of the 34 tasks and randomly sampled from them during evaluation.
> See the following table for some examples.
> More examples can be found in Tab. 6 in the updated paper.
>
> |                    Original                     |                     Generated                     |
> |:-----------------------------------------------:|:-------------------------------------------------:|
> |   "use the switch to turn off the light bulb"   |     "use the switch to stop the light source"     |
> | "slide the block that it falls into the drawer" | "Move the block ensuring it goes into the drawer" |
> | "take the red block and rotate it to the right" |        "Twist the red object to the right"        |
> |         "grasp and lift the blue block"         |      "grasp firmly and raise the blue block"      |
> |    "push the sliding door to the right side"    |   "Guide the sliding door to move on the right"   |
>
> For GR-1 and all the comparing baseline methods, we use the model trained on ABCD->D split for evaluation and test on the same sampled set of unseen language instructions.
> Results are shown in the following table.
>
> | Method     |   1   |   2   |   3   |   4   |   5   | Avg. Len. |
> |------------|:-----:|:-----:|:-----:|:-----:|:-----:|:---------:|
> | MT-R3M     | 0.512 | 0.249 | 0.106 | 0.040 | 0.017 |   0.92    |
> | RT-1       | 0.494 | 0.222 | 0.086 | 0.036 | 0.017 |   0.86    |
> | HULC       | 0.715 | 0.470 | 0.308 | 0.199 | 0.130 |   1.82    |
> | GR-1       | 0.764 | 0.555 | 0.381 | 0.270 | 0.196 |   2.17    |
>
> GR-1 outperforms all the comparing methods, improving the success rate from 71.5% to 76.4%.
> We hypothesize that this generalization capability attributes to 1) being exposed to diverse languages in the large video dataset during pre-training and 2) freezing the strong CLIP text encoder during the whole training process to retain its powerful text encoding capability.
> The strong performance on zero-shot generalization to unseen languages highlights GR-1's potential to be applied in daily life where human languages are diverse.
>
> [1] Mees, Oier, et al. "Calvin: A benchmark for language-conditioned policy learning for long-horizon robot manipulation tasks." IEEE Robotics and Automation Letters 7.3 (2022): 7327-7334.

---

> ### Author Response · Authors · 2023-11-23
> **Response to Reviewer H1qB [3/3]**
>
> **Q4**: ```It would be helpful to provide an ablation on the effect of different quantities of Ego4D pretraining datasets on downstream robot manipulation task performance. Also is there a (possibly smaller) subset of Ego4D dataset for pretraining that is especially helpful for downstream robot manipulation tasks?```
>
> **A4**:
> Thanks for your advice.
> We perform experiments on pre-training GR-1 on 1/4 and 1/2 of the full data used for pre-training in the paper respectively.
> We then finetune the pre-trained models on 10% data from ABCD-D split.
> Results are shown in the following table.
>
> | Pre-Training Data |   1   |   2   |   3   |   4   |   5   | Avg. Len. |
> |-------------------|:-----:|:-----:|:-----:|:-----:|:-----:|:---------:|
> | 1/4               | 0.655 | 0.395 | 0.247 | 0.144 | 0.077 |   1.52    |
> | 1/2               | 0.694 | 0.422 | 0.252 | 0.152 | 0.096 |   1.62    |
> | Full              | 0.778 | 0.533 | 0.332 | 0.218 | 0.139 |   2.00    |
>
> The above results show that the performance increases with the increase of pre-training data size.
> We are also interested in exploring whether there is a smaller subset of Ego4D dataset that is especially helpful for robot manipulation task.
> However, given the limited rebuttal time and computation resources, we are not able to perform systematic experiments to investigate this point.
> As indicated in the future work, we plan to explore more on this topic and compare the difference of pre-training on videos of any kind v.s. only videos that are more relevant to manipulation.
>
> **Q5**: ```The number of visual features per image from the MAE encoder can be large, and as a result, the current policy might not generalize to very long horizon tasks. It would be helpful to explore using e.g., Perceiver Resampler, to downsample the number of input image features.```
>
> **A5**:
> Yes, it is correct.
> And this is exactly what we have done in GR-1.
> We downsample the visual feature outputs from MAE with a perceiver resampler to 9 tokens.
> For more details, please refer to Fig. 2(c) and Sec. 3.2.1 in the paper.

---

### Official Review · Reviewer_Cjar · 2023-10-31

**Soundness:** 3 good
**Presentation:** 3 good
**Contribution:** 2 fair
**Rating:** 6
**Confidence:** 3

**Summary:**

The paper presents an approach for large-scale video pretraining for robot manipulation. The presented method, termed GR-1, first pre-trains a causal GPT-style transformer on actionless video data, and then finetunes the representation on a smaller number of robot demonstrations with actions. The key aspects of the approach are (1) using a GPT-style transformer for the embodied foundation model and (2) predicting the video tokens/generating future video frames as part of the training process. The approach is tested on the CALVIN simulation benchmark and on a real-world robot platform, outperforming the considered baselines.

**Strengths:**

* The general problem of making use of actionless human video data is of interest and importance to the research community.
* The problem is well-motivated and the literature review does a good job of contextualizing the paper in prior work.
* The paper is well-written and easy to follow.
* The figures are informative and effectively illustrate the benefits of the proposed approach.
* The experiments consider both simulation and real robot evaluation, as well as an ablation study, demonstrating GT-1's superior performance as compared to the considered baselines and support for GT-1's design choices.
* The discussion did a good job of describing the weaknesses and failure modes of the proposed method as well as the baselines.

**Weaknesses:**

* The literature review is missing a number of relevant works. The baselines considered are also not necessarily state-of-the-art. This year, a number of works have come out that would be prudent to compare against.
  * V-PTR: similar high-level motivation of using video-based, prediction-focused pre-training and then action-based finetuning. This should have likely served as a baseline for the proposed method.
    * [A] Bhateja, Chethan, et al. "Robotic Offline RL from Internet Videos via Value-Function Pre-Training." arXiv preprint arXiv:2309.13041 (2023).
  * Diffusion policy: diffusion policy has shown very good results in terms of multi-task, low-data regime performance.
    * [B] Chi, Cheng, et al. "Diffusion policy: Visuomotor policy learning via action diffusion." arXiv preprint arXiv:2303.04137 (2023).
    * [C] Ha, Huy, Pete Florence, and Shuran Song. "Scaling up and distilling down: Language-guided robot skill acquisition." arXiv preprint arXiv:2307.14535 (2023).
  * Pretrained video representations, such as R3M, VIP, and Voltron consider transformers, MAEs, and temporal video frames similarly to the proposed work. Taking a policy learning approach with a pre-trained representation such as the ones listed would have been another good option for a baseline.
    * [D] Nair, Suraj, et al. "R3M: A universal visual representation for robot manipulation." arXiv preprint arXiv:2203.12601 (2022).
    * [E] Ma, Yecheng Jason, et al. "VIP: Towards universal visual reward and representation via value-implicit pre-training." arXiv preprint arXiv:2210.00030 (2022).
    * [F] Karamcheti, Siddharth, et al. "Language-driven representation learning for robotics." arXiv preprint arXiv:2302.12766 (2023).
* Given above works, I am not sure that the statement "GR-1 for the first time shows that a unified GPT-style transformer, augmented with large-scale video generative pre-training is able to effectively generalize to multi-task visual robot manipulation." is accurate/precise.
* I found section 3.2.2 to be a bit confusing in describing the masked tokens. The OBS token is masked to predict the future video frames, is that correct? This sentence: "all the tokens, including the [OBS] tokens, can only attend to all the language and observation tokens but not the [OBS] in the past" was particularly unclear for me. Why specifically the tokens in the past? The notation of [OBS] and observation tokens adds to the confusion.
* In the experimental setup, I did not quite understand the actual size of the training set. My assumption is that GT-1 does not handle data both with and without language.
* It would be helpful to have some measure of statistical significance for the results (e.g., standard error) to understand whether the differences in performance are meaningful.
* Future work description should be more insightful/in-depth than 'combine more robot data and video data in training'.

Some typos and points of confusion are listed below:

1. Often 'causal' is typoed as 'casual'.
2. In Sec. 4.1, '[an] LED and a light bulb'?
3. In Sec. 4.1, 'datset'.
4. In Sec. 4.1, 'This highlight[s]' ... 'without collecting [a] large amount of data'.
5. The references should be proofread (e.g., to ensure the year is not entered twice, appropriate acronyms are capitalized (e.g., 'rl'), conference name formatting is consistent in terms of capitalization, etc.).

**Questions:**

1. How different is the D simulation setting from A, B, and C?
2. What is the $\Delta_t$ used in the experiments (e.g., 0.1 s)?
3. Why is only one action predicted at a time rather than a receding horizon style prediction? [B] found the latter to work well.
4. Is there multimodality in the distribution that can be modeled for the video frame prediction task?

---

> ### Author Response · Authors · 2023-11-23
> **Response to Reviewer Cjar [1/3]**
>
> Dear Reviewer,
>
> Thanks for your constructive comments. Below, we would like to address all the weaknesses and questions in details.
>
> **Q1**: ```The literature review is missing a number of relevant works. The baselines considered are also not necessarily state-of-the-art...V-PTR...Diffusion policy...Pretrained video representations, such as R3M, VIP, and Voltron consider transformers, MAEs, and temporal video frames similarly to the proposed work. Taking a policy learning approach with a pre-trained representation such as the ones listed would have been another good option for a baseline...```
>
> **A1**:
> We have added citations to all these relevant works in the paper.
> As for baseline comparison, V-PTR is a concurrent work and its code is not open-sourced.
> We added extensive experiments to compare with R3M in both simulation and the real world since it is also pre-trained on Ego4D dataset.
> - We first performed an experiment using a similar method proposed in the original R3M paper [1].
> We use the pre-trained R3M image encoder to encode the static image and the hand image of the current timestep into two image embeddings.
> In order to solve multiple tasks, we use the text encoder to encode the language instruction into text embedding.
> Text and image embeddings are passed through linear layers to align the dimension and then through an MLP to output actions.
> Following [1], the R3M image encoder is frozen during training.
> We denote this multi-task version of R3M as *MT-R3M-MLP*.
> - However, we find the performance of this method poor in both simulation and real robot experiments.
> We hypothesize the reason is that 1) the model does not have access to history 2) the number of training parameters is too small. We further implemented another version with R3M denoted as *MT-R3M*.
> Specifically, *MT-R3M* employs a gpt-style transformer to model the history information. The trainable parameters are the same as that in GR-1.
> - We also include Voltron results for reference. Here Voltron* means fine-tuning the entire model to get better results.
>
> Results on CALVIN ABCD split are shown in the following table. More results on CALVIN can be found in Tab. 1 in the paper.
>
> | Method     |   1   |   2   |   3   |   4   |   5   | Avg. Len. |
> |------------|:-----:|:-----:|:-----:|:-----:|:-----:|:---------:|
> | MT-R3M-MLP | 0.085 | 0.005 | 0.001 | 0.000 | 0.000 |   0.09    |
> | MT-R3M     | 0.752 | 0.527 | 0.375 | 0.258 | 0.163 |   2.08    |
> | Voltron    | 0.101 | 0.300 | 0.100 | 0.010 | 0.000 |   0.11    |
> | Voltron*   | 0.837 | 0.566 | 0.352 | 0.208 | 0.115 |   2.08    |
> | RT-1       | 0.844 | 0.617 | 0.438 | 0.323 | 0.227 |   2.45    |
> | HULC       | 0.889 | 0.733 | 0.587 | 0.475 | 0.383 |   3.06    |
> | GR-1       | 0.949 | 0.896 | 0.844 | 0.789 | 0.731 |   4.21    |
>
> We also compare with MT-R3M in two real robot experiments.
> Results on object transport experiment can be found in the following table.
>
> | Method | Seen Objects | Unseen Instances | Unseen Objects |
> |--------|:------------:|:----------------:|:--------------:|
> | RT-1   |     0.27     |       0.13       |      0.00      |
> | MT-R3M |     0.15     |       0.13       |      0.10      |
> | GR-1   |     0.79     |       0.73       |      0.30      |
>
> Results on articulated object manipulation experiment can be found in the following table.
>
> | Method | Average Success Rate |
> |--------|:--------------------:|
> | MT-R3M |         0.30         |
> | RT-1   |         0.35         |
> | GR-1   |         0.75         |
>
> On CALVIN benchmark, GR-1 outperforms MT-R3M in 1) multi-task learning on ABCD->D split 2) zero-shot generalization on ABC->D split 3) small dataset setting on 10% ABCD->D data.
> We also performed experiments on zero-shot generalization to unseen language instructions.
> And GR-1 achieves a success rate of 76.4% while MT-R3M achieves 51.2%.
> In real-robot experiments, GR-1 outperforms MT-R3M in both object transportation and articulated object manipulation.
> **These results highlight the strength of the video generative pre-training in GR-1 on visual manipulation learning.**
>
> [1] Nair, Suraj, et al. "R3M: A Universal Visual Representation for Robot Manipulation." arXiv preprint arXiv:2203.12601 (2022).

---

> ### Author Response · Authors · 2023-11-23
> **Response to Reviewer Cjar [2/3]**
>
> **Q2**: ```Given above works, I am not sure that the statement "GR-1 for the first time shows that a unified GPT-style transformer, augmented with large-scale video generative pre-training is able to effectively generalize to multi-task visual robot manipulation." is accurate/precise.```
>
> **A2**:
> We are sorry for causing the confusion. What differs GR-1 from the mentioned relevant works is 1) its pre-training objective--*video generative pre-training* and 2) its model architecture--*GPT style transformer*.
> - V-PTR, concurrent to our work, leverages *video latent dynamics modeling* to learn value functions and refine the visual features by offline RL on robot data.
> - VIP uses large-scale human videos (Ego4D) to perform value-implicit pre-training which is able to provide visual reward and representation for downstream robotic tasks. VIP can be viewed as an *implicit time contrastive objective* as indicated in the paper.
> - The two papers on Diffusion Policy did not exploit video pre-training.
> - R3M and MVP also uses Ego4D dataset for pre-training. The pre-training objective for R3M and MVP are *contrastive learning* and *masked image reconstruction* respectively.
>
> While some of the mentioned relevant works use video for pre-training, none performed *generative pre-training* for videos.
> While some works leverages transformers as the model architecture, none of them uses GPT-style transformers.
> Therefore, we believe *GR-1 is the first work which shows that a unified GPT-style transformer, augmented with large-scale video generative pre-training, is able to effectively generalize to multi-task visual robot manipulation*. Furthermore, inspired by the successes of the GPTs in language and multi-modal applications, we hope that GR-1 can modestly contribute to the expansion of these achievements.
>
> **Q3**: ```I found section 3.2.2 to be a bit confusing in describing the masked tokens. The OBS token is masked to predict the future video frames, is that correct? This sentence: "all the tokens, including the [OBS] tokens, can only attend to all the language and observation tokens but not the [OBS] in the past" was particularly unclear for me. Why specifically the tokens in the past? The notation of [OBS] and observation tokens adds to the confusion.```
>
> **A3**:
> We are sorry for causing the confusion.
> GR-1 follows the causal attention mechanism typically used in generative pre-trained models, except that all the [ACT] and [OBS] tokens are masked.
> The [OBS] tokens are masked to predict future video frames; the [ACT] tokens are masked to predict the actions.
> During pre-training, a token can attend to all the tokens in the previous positions except the [OBS] tokens;
> during finetuning, all the tokens can attend to all the tokens in the previous positions except the [ACT] and [OBS] tokens. We have updated the writing of this part for clarification in Sec. 3.2.2 of the updated paper.
>
> **Q4**: ```In the experimental setup, I did not quite understand the actual size of the training set. My assumption is that GT-1 does not handle data both with and without language.```
>
> **A4**:
> The training set of ABCD->D split contains 22,966 trajectories with language annotations.
> It is true that GR-1 does not handle data without language.
> In the future, we plan to extend GR-1 to handle both types of data and make use of more data in training.
>
> **Q5**: ```It would be helpful to have some measure of statistical significance for the results (e.g., standard error) to understand whether the differences in performance are meaningful.```
>
> **A5**:
> Thanks for your advice! In the following, we show the mean and standard deviation of three seeded runs of GR-1 on ABCD->D and ABC->D splits.
>
> | Splits |       1       |       2       |       3       |       4       |       5       |  Avg. Len.   |
> |:------:|:-------------:|:-------------:|:-------------:|:-------------:|:-------------:|:------------:|
> |  ABCD  | 0.949 (0.004) | 0.896 (0.008) | 0.844 (0.003) | 0.789 (0.008) | 0.731 (0.003) | 4.21 (0.02)  |
> |  ABC   | 0.854 (0.014) | 0.712 (0.014) | 0.596 (0.017) | 0.497 (0.020) | 0.401 (0.016) | 3.06 (0.079) |
>
> As shown in the table, the standard deviation is very small, indicating that the performance of GR-1 is stable.

---

> ### Author Response · Authors · 2023-11-23
> **Response to Reviewer Cjar [3/3]**
>
> **Q6**: ```Future work description should be more insightful/in-depth than 'combine more robot data and video data in training'.```
>
> **A6**:
> We updated the future work discussion in the paper for more insightful and in-depth discussion.
> *By incorporating large-scale video data, we showcase that GR-1 is able to perform robustly in scenes which are disturbed heavily from those in the training data.
> More importantly, GR-1 is able to generalize to unseen object instances and categories in a zero-shot manner.
> In the future, we hope to combine video data both with and without language in training to further enhance the robustness and generalization capability of GR-1.
> Also, we want to explore the difference of pre-training on videos of any kind v.s. only videos that are more relevant to manipulation.
> In addition, we plan to scale up the robot data by increasing both the number of robot trajectories in diverse environments and the number of manipulation skills.*
>
> **Q7**: ```Some typos and points of confusion are listed below: Often 'causal' is typoed as 'casual'...```
>
> **A7**:
> We appreciate the reviewer for pointing out these typos.
> In the updated paper, we have corrected all the mentioned typos in the paper and the reference.
>
> **Q8**: ```How different is the D simulation setting from A, B, and C?```
>
> **A8**:
> As shown in Fig. 4 in the paper, the four environments are different in table textures and colors.
> Also, the positions of the sliding door, LED, bulb, light switch, and button are different across the four environments.
> These large variations require the model to possess strong semantic understanding capabilities and accurate visual-motor control.
> GR-1 outperforms the comparing baseline methods by a large margin in the zero-shot generalization setting, improving the averaged task success rate from 53.3% to 85.4%.
> This shows the strong zero-shot generalization capability of GR-1 to unseen environments.
>
> **Q9**: ```What is the $\Delta t$ used in the experiments (e.g., 0.1 s)?```
>
> **A9**:
> The $\Delta t$ in the paper specifies the target image for a prediction in the sequence.
> In pre-training, we sampled frames in which the duration between adjacent frames is 1/3s.
> GR-1 is trained to predict the next frame in the sampled sequence, which is 1/3s in the future.
> For CALVIN data and real robot data, GR-1 is trained to predict the 3rd frame in the future as the robot data is denser compared to the down-sampled Ego4D videos.
>
> **Q10**: ```Why is only one action predicted at a time rather than a receding horizon style prediction? [B] found the latter to work well.```
>
> **A10**:
> We appreciate the reviewer for bringing up the receding horizon prediction.
> The proposed method GR-1 is a principled method and can be easily extended to support receding horizon control (RHC) by attaching a policy head which is able to predict actions for multiple steps.
> During the limited rebuttal time, we work diligently to implement an RHC-style GR-1 by changing the prediction of the action head from predicting one action to multiple actions.
> Results on CALVIN ABCD split are shown in the following table.
>
> | Method      |   1   |   2   |   3   |   4   |   5   | Avg. Len. |
> |-------------|:-----:|:-----:|:-----:|:-----:|:-----:|:---------:|
> | GR-1        | 0.949 | 0.896 | 0.844 | 0.789 | 0.731 |   4.21    |
> | GR-1 w/ RHC | 0.950 | 0.902 | 0.857 | 0.816 | 0.761 |   4.29    |
>
> We plan to investigate more on this point in the future.
>
> **Q11**: ```Is there multimodality in the distribution that can be modeled for the video frame prediction task?```
>
> **A11**:
> Currently, the video prediction of GR-1 does not support multi-modal prediction given that it uses an L2 loss to reconstruct the target frames.
> However, GR-1 can be easily extended to involve multi-modality in video prediction.
> One possible approach is to attach a VAE/VQ-GAN decoder for predicting the image.
> And the video prediction can be trained by the corresponding generative loss to support multi-modality in video prediction.
> We will investigate more on this point in the future.

---

### Official Review · Reviewer_SaZc · 2023-10-31

**Soundness:** 2 fair
**Presentation:** 3 good
**Contribution:** 2 fair
**Rating:** 5
**Confidence:** 4

**Summary:**

The paper develops an approach for multi-task visual robotic manipulation by pre-training for video prediction on non-robotic datasets, and then fine-tuning on robot data. The main contribution of the paper is to demonstrate that pre-training for the task of video prediction on human datasets enables sample-efficient fine-tuning for manipulation, with some generalization across tasks. The experiments involve manipulation tasks on a simulation environment, and pick/place tasks in the real world.

**Strengths:**

- The task of video prediction on non-robot datasets for as a pre-training step for downstream manipulation is interesting, and novel to the best of my understanding. Importantly, this pre-training task can ingest all language-annotated video datasets, can hence can be potentially scaled to diverse videos on the web, beyond the Ego4D dataset in the paper.

- Three external baselines are implemented for comparison, and experiments show some evidence of real-world manipulation, which strengthens the claims of the paper.

- The vision transformer architecture is without significant bells and whistles, and has standard GPT-style encoder-decoder structure, making it easy to implement and train. In my understanding, the fine-tuning on robot data is very similar to the pre-training step in terms of training details, and doesn't require additional considerations.

- Overall, the paper is well-written, easy to understand, and the approach is clearly described.

**Weaknesses:**

- The results doesn't quite succeed in showing any generalization benefits of pre-training for video prediction on diverse data. This seems to be the main claim of the paper, and requires more thorough experiments for validation. The simulation environments seem to have all the tasks in a similar table-top setting, with very little scene variation across tasks. The zero-shot result is interesting, but it is shown for only a single task.

- The real-world experiments are in very simple pick and place tasks. The Ego4D videos used for pre-training contain such rich skills like articulated object manipulation, scooping, pouring etc. and so for properly showing the benefits of pre-training it is important to demonstrate that the approach can indeed tackle some of the interesting contact rich tasks that involve motions/skills beyond pick and place. In addition, the generalization results are limited to minor distractor and background variations, without variations in the task objects themselves. For reference, several recent robotics papers tackle such tasks and generalizations [1,2]

- The baselines are missing all relevant papers that also pre-train on human videos / image datasets. Many of these are cited in the related works, but some comparison is necessary to actually demonstrate that pre-training for video prediction has benefits compared to pre-training for other objects like masked prediction, contrastive learning etc. (which is a claim of the paper). For example, a multi-task version of R3M [3] can be a relevant baseline that pre-trains on Ego4D but to learn visual representations, and then fine-tunes with behavior cloning.

[1] Walke, Homer, Kevin Black, Abraham Lee, Moo Jin Kim, Max Du, Chongyi Zheng, Tony Zhao et al. "Bridgedata v2: A dataset for robot learning at scale." arXiv preprint arXiv:2308.12952 (2023).

[2] Bharadhwaj, Homanga, Jay Vakil, Mohit Sharma, Abhinav Gupta, Shubham Tulsiani, and Vikash Kumar. "Roboagent: Generalization and efficiency in robot manipulation via semantic augmentations and action chunking." arXiv preprint arXiv:2309.01918 (2023).

[3] Suraj Nair, Aravind Rajeswaran, Vikash Kumar, Chelsea Finn, and Abhinav Gupta. R3M: A universal visual representation for robot manipulation. In 6th Annual Conference on Robot Learning,
2022.

**Questions:**

Please refer to the weaknesses above for reference.

- Is it possible to show generalization results (similar in line to the zero-shot results) but for more tasks and variations within tasks, like different scenes, different objects etc.? This is expecially important because the current simulation benchmark seems saturated with 88% baseline performance, and only marginal improvement to 94% with the proposed approach.

- For the real-world experiments, is it possible to show benefits in tasks beyond pick/place and those that frequently occur in the pre-training Ego4D dataset like articulated object manipulation, scooping, pouring etc.?

- Why have ABC-> D style result instead of training on all the available tasks (except the held-out one)? I might be missing something here, so would appreciate a clarification, but it seems to me that the zero-shot result in the paper doesn't seem to be training a unified model across all but held out tasks, which seems a bit odd.

- Is it possible to have comparisons to prior works that also train on datasets similar/identical to Ego4D but have different pre-training objectives? (please refer to weaknesses above for details)

- Is it possible to show qualitative results for the video-prediction (similar to Fig 6 and Fig 11) but for more real-world evaluations? If possible it will be helpful to show some *random* generations to get a sense of how much

- Is there no forgetting in the pre-training followed by fine-tuning regime (as opposed to co-training)? My sense is that after fine-tuning on the robot dataset, the information acquired from pretraining will be largely lost due to domain shift in the two datasets. One simple way to test this is to look at video generation results of the model on Ego4D clips, just after pre-training, and after pre-training+finetuning. My sense is that the latter will be much worse than the former. I am curious if the authors considered any co-training strategy, where for fine-tuning, the dataset consists of some clips from the pre-training dataset as well as the robot data?

---

> ### Author Response · Authors · 2023-11-23
> **Response to Reviewer SaZc [1/4]**
>
> Dear Reviewer,
>
> Thanks for your constructive comments. Below, we would like to address all the weaknesses and questions in details.
>
> **Response to Weaknesses**
>
> **Q1**: ```The results doesn't quite succeed in showing any generalization benefits of pre-training for video prediction on diverse data. This seems to be the main claim of the paper, and requires more thorough experiments for validation. ...The zero-shot result is interesting, but it is shown for only a single task.```
>
> **A1**:
> Regarding the simulation environment, we believe there is a misunderstanding here.
> In the experiment on zero-shot generalization to unseen scenes (ABC->D split), we follow the evaluation protocol in [CALVIN](http://calvin.cs.uni-freiburg.de) [1].
> We train GR-1 on data collected from environment A, B, and C and evaluate it in environment D which is unseen during training.
> The training data contains all the 34 tasks collected from A, B, and C.
> And the evaluation is performed on all the 34 tasks, instead of one single task, in environment D.
> As shown in Fig. 4 in the paper, the four environments are different in table textures and colors.
> Also, the positions of the sliding door, LED, bulb, light switch, and button are different across the four environments.
> These large variations require the model to possess strong semantic understanding capabilities and accurate visual-motor control.
> GR-1 outperforms the comparing baseline methods by a large margin, improving the averaged task success rate from 53.3% to 85.4%.
> Please refer to Sec. 4.1 in the paper for more details.
>
> In addition, we have added experiments on zero-shot generalization to *unseen languages*. To do so, we use GPT-4 to generate 50 synonymous instructions for each of the 34 tasks and randomly sampled from them during evaluation.
> GR-1 outperforms all the comparing baseline methods.
> This demonstrates the potential of GR-1 being applied to daily scenarios where generalization to unseen languages is essential given that human languages are diverse.
> Results can be found in the following table.
> For the details of MT-R3M, please see the response to **Q3** and more details can be found in Sec. 4.1 in the updated paper.
>
> | Method |   1   |   2   |   3   |   4   |   5   | Avg. Len. |
> |--------|:-----:|:-----:|:-----:|:-----:|:-----:|:---------:|
> | RT-1   | 0.494 | 0.222 | 0.086 | 0.036 | 0.017 |   0.86    |
> | HULC   | 0.715 | 0.470 | 0.308 | 0.199 | 0.130 |   1.82    |
> | MT-R3M | 0.512 | 0.249 | 0.106 | 0.040 | 0.017 |   0.92    |
> | GR-1   | 0.764 | 0.555 | 0.381 | 0.270 | 0.196 |   2.17    |
>
> Futhermore, in real-robot experiments, we added experiments on *unseen object manipulation*.
> Specifically, we test GR-1 on transporting unseen objects of seen categories and unseen objects of unseen categories.
> GR-1 outperforms the comparing baseline methods by a large margin.
> More details can be found in Sec. 4.2 in the paper.
>
> We believe GR-1's strong results on generalization to unseen environments, unseen languages, and unseen objects together show the advantages of pre-training on diverse video data.
>
> [1] Mees, Oier, et al. "Calvin: A benchmark for language-conditioned policy learning for long-horizon robot manipulation tasks." IEEE Robotics and Automation Letters 7.3 (2022): 7327-7334.

---

> ### Author Response · Authors · 2023-11-23
> **Response to Reviewer SaZc [2/4]**
>
> **Q2**: ```The real-world experiments are in very simple pick and place tasks. The Ego4D videos used for pre-training contain such rich skills like articulated object manipulation, scooping, pouring etc. and so for properly showing the benefits of pre-training it is important to demonstrate that the approach can indeed tackle some of the interesting contact rich tasks that involve motions/skills beyond pick and place. In addition, the generalization results are limited to minor distractor and background variations, without variations in the task objects themselves. For reference, several recent robotics papers tackle such tasks and generalizations [1,2]```
>
> **A2**:
> We have added two real-robot experiments in the paper to address this point.
>  - We conducted an experiment on articulated object manipulation.
> We collected 2800+ trajectories of opening and closing a drawer and train GR-1 on these two tasks.
> We compare with two baseline methods including RT-1 and a multi-task version of R3M (MT-R3M).
> For the details of MT-R3M, please see the response to **Q3** and Sec. 4.1 in the paper.
> Results are shown in the following table.
> GR-1 outperforms the comparing baseline methods in articulated object manipulation.
>
>     | Method | Average Success Rate |
>     |--------|:--------------------:|
>     | MT-R3M |         0.30         |
>     | RT-1   |         0.35         |
>     | GR-1   |         0.75         |
>
> - We performed experiments on unseen objects to address variations in the task objects.
> Specifically, we evaluate GR-1 on transporting unseen object instances of seen categories and unseen objects of unseen categories.
> For the unseen instances, we evaluate on a novel set of broccoli, bell pepper, and eggplant which are unseen in the robot data.
> For the unseen objects of unseen categories, we evaluate on a tomato and a yellow peach which are unseen in the language instructions of the robot data.
> Results are shown in the following table and more details can be found in Sec 4.1 of the paper.
> GR-1 outperforms RT-1 and MT-R3M in transporting both unseen instances and unseen objects of unseen categories, showing strong generalization capabilities to novel objects.
>
>     | Method | Unseen Instances | Unseen Categories |
>     |--------|:----------------:|:--------------:|
>     | RT-1   |      0.13       |     0.00      |
>     | MT-R3M |      0.13       |     0.10      |
>     | GR-1   |      0.73       |     0.30      |

---

> ### Author Response · Authors · 2023-11-23
> **Response to Reviewer SaZc [3/4]**
>
> **Q3**: ```The baselines are missing all relevant papers that also pre-train on human videos / image datasets. Many of these are cited in the related works, but some comparison is necessary to actually demonstrate that pre-training for video prediction has benefits compared to pre-training for other objects like masked prediction, contrastive learning etc. (which is a claim of the paper). For example, a multi-task version of R3M [3] can be a relevant baseline that pre-trains on Ego4D but to learn visual representations, and then fine-tunes with behavior cloning.```
>
> **A3**:
> We have added experiments to compare with R3M in the updated paper.
> We use the pre-trained R3M image encoder to encode the static image and the hand image of the current timestep into two image embeddings.
> In order to solve multiple tasks, we use the text encoder to encode the language instruction into text embedding.
> These embeddings are passed through linear layers to align the dimension and then through an MLP to output actions.
> Follwing [2], the R3M image encoder is frozen during training.
> We denote this multi-task version of R3M as *MT-R3M-MLP*.
> However, we find the performance of this method poor.
> We hypothesize the reason is that 1) the model does not have access to history 2) the number of training parameters is too small.
> We further implemented another version, denoted as *MT-R3M*, using R3M as the image encoder.
> MT-R3M employs a GPT-style transformer to model the history information. The number of trainable parameters is the same as that in GR-1.
> Results on CALVIN ABCD->D split are shown in the following table.
> More results on CALVIN benchmark can be found in Tab. 1 of the updated paper.
>
> | Method     |   1   |   2   |   3   |   4   |   5   | Avg. Len. |
> |------------|:-----:|:-----:|:-----:|:-----:|:-----:|:---------:|
> | MT-R3M-MLP | 0.085 | 0.005 | 0.001 | 0.000 | 0.000 |   0.09    |
> | MT-R3M     | 0.752 | 0.527 | 0.375 | 0.258 | 0.163 |   2.08    |
> | RT-1       | 0.844 | 0.617 | 0.438 | 0.323 | 0.227 |   2.45    |
> | HULC       | 0.889 | 0.733 | 0.587 | 0.475 | 0.383 |   3.06    |
> | GR-1       | 0.949 | 0.896 | 0.844 | 0.789 | 0.731 |   4.21    |
>
> We also compare R3M in real robot experiments as shown in the response to **Q2**.
> On CALVIN benchmark, GR-1 outperforms MT-R3M in 1) multi-task learning on ABCD->D split 2) zero-shot generalization on ABC->D split and 3) small dataset setting on 10% ABCD->D data.
> We also performed experiments on zero-shot generalization to unseen language instructions.
> And GR-1 achieves a success rate of 76.4% while MT-R3M achieves 51.2%.
> In real-robot experiments, GR-1 outperforms MT-R3M in both object transportation and articulated object manipulation.
>
> **These results all highlight the strength of the video generative pre-training in GR-1 on visual manipulation learning.**
>
> [2] Nair, Suraj, et al. "R3M: A Universal Visual Representation for Robot Manipulation." arXiv preprint arXiv:2203.12601 (2022).

---

> ### Author Response · Authors · 2023-11-23
> **Response to Reviewer SaZc [4/4]**
>
> **Response to Questions**
>
> **Q4**: ```Is it possible to show generalization results (similar in line to the zero-shot results) but for more tasks and variations within tasks, like different scenes, different objects etc.? This is expecially important because the current simulation benchmark seems saturated with 88% baseline performance, and only marginal improvement to 94% with the proposed approach.```
>
> **A4**:
> As discussed in the response to **Q1**, we believe there is a misunderstanding here.
> The simulation experiment on CALVIN ABC->D split evaluates GR-1 on 34 tasks in an unseen environment.
> And GR-1 achieves a success rate of 85.4% while the best baseline method achieves 53.3%.
> Moreover, we added real robot experiments on zero-shot generalization to unseen objects in the paper as shown in the response to **Q2**.
> GR-1 outperforms the comparing baseline methods by a large margin.
> All these results on generalization to different scenes and objects showcase the advantage of GR-1's generalization capability.
>
> **Q5**: ```For the real-world experiments, is it possible to show benefits in tasks beyond pick/place and those that frequently occur in the pre-training Ego4D dataset like articulated object manipulation, scooping, pouring etc.?```
>
> **A5**:
> As discussed in the response to **Q2**, we have added experiments on articulated object manipulation.
> Specifically, we performs experiments on opening and closing a drawer.
> GR-1 outperforms RT-1 and MT-R3M on the averaged task success rate.
> In the future, we plan to continue to scale up the number of manipulation tasks, including pouring, scooping, and more artiuclated object manipulation.
>
> **Q6**: ```Why have ABC-> D style result instead of training on all the available tasks (except the held-out one)? I might be missing something here, so would appreciate a clarification, but it seems to me that the zero-shot result in the paper doesn't seem to be training a unified model across all but held out tasks, which seems a bit odd.```
>
> **A6**:
> As discussed in the response to **Q1** and **Q4**, in zero-shot generalization experiments on ABC->D split, we train GR-1 on all the available 34 tasks with data collected from environment A, B, and C.
> The evaluation is conducted in environment D, which is unseen in the training data, on all the 34 tasks.
>
> **Q7**: ```Is it possible to have comparisons to prior works that also train on datasets similar/identical to Ego4D but have different pre-training objectives? (please refer to weaknesses above for details)```
>
> **A7**:
> Thanks for your advice. We have added comparison with R3M, which is also pre-trained on Ego4D but with a contrastive learning objective, in both CALVIN simulation and the real world as discussed in the response to **Q3**.
> GR-1 outperforms MT-R3M in all the settings on CALVIN benchmark and all the settings in the real robot experiments.
> More results on CALVIN benchmark and real robots can be found in Sec. 4.1 and 4.2 in the paper.
>
> **Q8**: ```Is it possible to show qualitative results for the video-prediction (similar to Fig 6 and Fig 11) but for more real-world evaluations? If possible it will be helpful to show some random generations to get a sense of how much```
>
> **A8**:
> We have included more qualitative results for video prediction in the real world in the updated paper.
> Please refer to Fig. 11 and Fig. 6 in the paper.
>
> **Q9**: ```Is there no forgetting in the pre-training followed by fine-tuning regime (as opposed to co-training)? My sense is that after fine-tuning on the robot dataset, the information acquired from pretraining will be largely lost due to domain shift in the two datasets. One simple way to test this is to look at video generation results of the model on Ego4D clips, just after pre-training, and after pre-training+finetuning. My sense is that the latter will be much worse than the former. I am curious if the authors considered any co-training strategy, where for fine-tuning, the dataset consists of some clips from the pre-training dataset as well as the robot data?```
>
> **A9**:
> Here we show the prediction loss on the validation set of Ego4D for the model before and after finetuning.
> Besides, we also compare the loss on the validation set of CALVIN.
> Results are shown in the following table.
>
> | Model | Loss on Ego4D | Loss on CALVIN |
> | ------ | :---: | :---: |
> | Before Finetuning  | 0.57 | 0.45 |
> | After Finetuning   | 1.17 | 0.42 |
>
> We appreciate the reviewer for bringing up co-training.
> We are also curious on whether it can alleviate the forgetting and further enhance the performance on visual manipulation learning.
> We plan to investigate it in future work.

---

### Official Review · Reviewer_QyTe · 2023-11-04

**Soundness:** 1 poor
**Presentation:** 3 good
**Contribution:** 2 fair
**Rating:** 3
**Confidence:** 4

**Summary:**

The authors introduce GR-1, a GPT-style decoder-only transformer for language-conditioned robot manipulation. The model predicts both actions and future observations; the authors pretrain on Ego4D predicting future observations, followed by finetuning on robot data predicting both future observations and actions. The resulting model shows strong performance on the CALVIN benchmark as well as real-world language-conditioned robot manipulation on several pick-and-place tasks.

**Strengths:**

The paper is well-written, well-presented, and easy to follow. The idea is simple and easy to understand, which is a good thing. While the basic idea of training a GPT-style transformer in this manner is not exactly brilliant or surprising, it hasn't been done in prior work (as far as I know), and the design of the model is principled and well-executed. The CALVIN results are strong, improving significantly on the state-of-the-art.

**Weaknesses:**

The value of GPT-style video pretraining is the core narrative of this paper, and the primary way that the authors claim GR-1 improves on prior work. Given that, I would say the main weaknesses of this paper are related to a lack of sufficient evidence and discussion surrounding this claim.

- Training details/hyperparameters/etc are missing for the Ego4D pretraining phase, which should definitely be included given their importance to the method.
- The real robot experiments are a bit weak. There are not very many tasks, they are fairly simple, and the tested level of generalization is not very significant. Breaking down the tasks into separate pick and place phases provides some inherent supervision that makes the tasks  easier; I would have liked to see slightly more end-to-end tasks (e.g., "put the broccoli on the plate"). Given how broad the pretraining dataset is, if it really does help, it seems like better zero-shot generalization should be possible: e.g., slightly different objects not found in the robot data.
- The baselines are severely lacking. For example, the authors claim that "compared with VIMA, our method is capable of dealing with much more complex 6-DoF long-horizon language-conditioned robot manipulation tasks". A statement like that should be supported by a comparison to some sort of VIMA-like method (i.e., an encoder-decoder architecture with cross-attention). There is also *tons* of prior work on video pretraining for robotics, much of which is cited in Section 2.3: e.g., R3M, VIP, MVP, and many more. Many of these also pretrain on Ego4D and finetune on robot data. It is impossible to evaluate the strength of GR-1 without comparison to these other methods.

I would be willing to increase my score if more comparisons were added in both the simulation and the real world.

**Questions:**

- For the Ego4D pretraining, did you use the entire dataset, or some more relevant subset (e.g., hands and objects)?
- Why was $\Delta t = 1$ used during pretraining? If I understand correctly, this means that the next frame in Ego4D was always predicted, which is temporally very close considering that Ego4D videos are fairly high FPS.
- Why is GR-1 w/o pretraining so much better than the baselines in the CALVIN zero-shot setting?
-  "283M parameters, in which 46M of them are trainable" -- to clarify, this means that the majority of the parameters are the frozen MAE and text encoder parameters?
- What is the control frequency of the real robot during the execution of the policy?

---

> ### Author Response · Authors · 2023-11-23
> **Response to Reviewer QyTe [1/3]**
>
> Dear Reviewer,
>
> Thanks for your constructive comments. Below, we would like to address all the weaknesses and questions in details.
>
> **Response to Weaknesses**
>
> **Q1**: ```Training details/hyperparameters/etc are missing for the Ego4D pretraining phase, which should definitely be included given their importance to the method.```
>
> **A1**:
> We have supplemented more training details and hyperparameters in Appendix A.1. In pre-training, we sample equally-spaced frames from video clips to train GR-1.
> The duration between consecutive frames is 1/3 seconds to ensure that they have sufficient visual difference. At each timestep, GR-1 is trained to predict the image at the next timestep in the sampled frame sequence. The training hyperparameters are shown as follows.
>
> | hyperparameters        | Pre-training |  Finetuning  |
> |------------------------|:------------:|:------------:|
> | batch size             |     1024     |     512      |
> | learning rate          |    3.6e-4    |     1e-3     |
> | dropout                |     0.1      |     0.1      |
> | optimizer              |    AdamW     |    AdamW     |
> | learning rate schedule | cosine decay | cosine decay |
> | warmup epochs          |      5       |      1       |
> | total epochs           |      50      |      20      |
>
>
> **Q2**: ```...I would have liked to see slightly more end-to-end tasks (e.g., "put the broccoli on the plate"). Given how broad the pretraining dataset is, if it really does help, it seems like better zero-shot generalization should be possible: e.g., slightly different objects not found in the robot data.```
>
> **A2**:
> We perform more real-robot experiments to showcase the capability of GR-1 on end-to-end tasks and zero-shot generalization to unseen objects.
> - We added the end-to-end object transportation experiment accordingly in Sec. 4.2.1. Example language instructions include "put the broccoli onto the plate" and "put the bell pepper onto the desk".
> Similar to the setting in the first submission, testing is performed in 1) an environment that only contains objects seen during training and 2) environments that contain distractors and background variation.
> - In addition, we add a more challenging setting where the objects are all unseen in the robot training data.
> In this setting, there are five objects, i.e. a bell pepper, a broccoli, an eggplant, a tomato, and a yellow peach.
> While the categories of three objects are seen in robot training data, they are different instances. The categories of the other two are unseen.
> This setting helps us evaluate GR-1's generalization capability for both unseen instances and unseen categories.
>
> In all settings, GR-1 outperforms the comparing baseline methods by a large margin.
> Results of object transportation experiments are shown in the following table.
> More details can be found in Sec. 4.2.1 in the updated paper.
>
> | Method | Seen Objects | Unseen Instances | Unseen Categories |
> |--------|:------------:|:----------------:|:-----------------:|
> | RT-1   | 0.27         | 0.13             | 0.00              |
> | MT-R3M | 0.15         | 0.13             | 0.10              |
> | GR-1   | 0.79         | 0.73             | 0.30              |
>
> Furthermore, we added experiments on articulated object manipulation in Sec. 4.2.2.
> Results of articulated object manipulation experiments are shown in the following table.
> GR-1 outperforms the comparing baseline methods in manipulating articulated objects.
>
> | Method | Success Rate |
> |--------|:------------:|
> | RT-1   | 0.35         |
> | MT-R3M | 0.30         |
> | GR-1   | 0.75         |

---

> ### Author Response · Authors · 2023-11-23
> **Response to Reviewer QyTe [2/3]**
>
> **Q3**: ```The baselines are severely lacking...There is also tons of prior work on video pretraining for robotics, much of which is cited in Section 2.3: e.g., R3M, VIP, MVP, and many more. Many of these also pretrain on Ego4D and finetune on robot data. It is impossible to evaluate the strength of GR-1 without comparison to these other methods.```
>
> **A3**:
> Thanks for your advice. We have included comparison with R3M since it is also pre-trained on Ego4D.
> We use the pre-trained R3M image encoder to encode the static image and the hand image into two image embeddings.
> In order to solve multiple tasks, we use the text encoder to encode the language instruction into text embedding.
> These embeddings are passed through linear layers to align the dimension and then through an MLP to output actions.
> We denote this multi-task version of R3M as *MT-R3M-MLP*.
> However, we find the performance of this method poor.
> We hypothesize the reason is that 1) the model does not have access to history and 2) the number of training parameters is too small.
> We further implemented another version, denoted as *MT-R3M*, using R3M as the image encoder.
> MT-R3M employs a GPT-style transformer to model the history information. The number of trainable parameters is the same as that in GR-1.
> Results are shown in the following.
> - On CALVIN benchmark, GR-1 outperforms MT-R3M in 1) multi-task learning on ABCD->D split 2) zero-shot generalization on ABC->D split 3) small dataset setting on 10% ABCD->D data. The following table shows the results trained on ABCD->D split. More results on CALVIN can be found in Tab. 1 in the paper.
>
>     | Method     |   1   |   2   |   3   |   4   |   5   | Avg. Len. |
>     |------------|:-----:|:-----:|:-----:|:-----:|:-----:|:---------:|
>     | MT-R3M-MLP | 0.085 | 0.005 | 0.001 | 0.000 | 0.000 |   0.09    |
>     | MT-R3M     | 0.752 | 0.527 | 0.375 | 0.258 | 0.163 |   2.08    |
>     | RT-1       | 0.844 | 0.617 | 0.438 | 0.323 | 0.227 |   2.45    |
>     | HULC       | 0.889 | 0.733 | 0.587 | 0.475 | 0.383 |   3.06    |
>     | GR-1       | 0.949 | 0.896 | 0.844 | 0.789 | 0.731 |   4.21    |
>
> - We also compare with MT-R3M in real robot experiments. Results on object transportation and articulated object manipulation can be found in response to **Q2**. GR-1 outperforms MT-R3M in both object transportation and articulated object manipulation.
>
> - We also performed experiments on zero-shot generalization to unseen language instructions. To do so, we use GPT-4 to generate 50 synonymous instructions for each of the 34 tasks and randomly sampled from them during evaluation.
> See Tab. 6 in the paper for some examples.
> For GR-1 and all the comparing baseline methods, we use the model trained on ABCD->D split for evaluation and test on the same set of sampled unseen language instructions.
> Results are shown in the following table.
>
>     | Method     |   1   |   2   |   3   |   4   |   5   | Avg. Len. |
>     |------------|:-----:|:-----:|:-----:|:-----:|:-----:|:---------:|
>     | MT-R3M     | 0.512 | 0.249 | 0.106 | 0.040 | 0.017 |   0.92    |
>     | RT-1       | 0.494 | 0.222 | 0.086 | 0.036 | 0.017 |   0.86    |
>     | HULC       | 0.715 | 0.470 | 0.308 | 0.199 | 0.130 |   1.82    |
>     | GR-1       | 0.764 | 0.555 | 0.381 | 0.270 | 0.196 |   2.17    |
>
>     The performance of all the methods drops when evaluated on unseen language instructions.
>     GR-1 outperforms all the comparing baseline methods.
>     We hypothesize that this generalization capability attributes to being exposed to diverse languages in the large video dataset during pre-training.
>
> **These results all highlight the strength of the video generative pre-training in GR-1 on visual manipulation learning.**

---

> ### Author Response · Authors · 2023-11-23
> **Response to Reviewer QyTe [3/3]**
>
> **Response to Questions**
>
> **Q4**: ```For the Ego4D pretraining, did you use the entire dataset, or some more relevant subset (e.g., hands and objects)?```
>
> **A4**:
> We only use the video-text paired video clips which is annotated by Ego4D teams for GR-1 pre-training. The entire Ego4D dataset contains more videos without text annotations. In the future, we are interested in comparing the performance of pre-training on videos of any kind v.s. only videos that are more relevant to manipulation (e.g., relevant subset of hands and objects).
>
> **Q5**: ```Why was $\Delta t=1$ used during pretraining? If I understand correctly, this means that the next frame in Ego4D was always predicted, which is temporally very close considering that Ego4D videos are fairly high FPS.```
>
> **A5**:
> Thanks for pointing out this confusion.
> We sampled image sequences from the video with an equal space of 1/3s.
> That is, the duration between two adjacent sampled frames is 1/3s.
> $\Delta t=1$ means that the target observation image of the prediction at i-th timestep in the sampled sequence is the image at i+1-th timestep.
> That is, the target image is the frame at 1/3s in the future, which is the next 10-th frame (the FPS of Ego4D is 30), instead of the immediate next first frame.
> We have clarified this point in Appendix A.1.
>
> **Q6**: ```Why is GR-1 w/o pretraining so much better than the baselines in the CALVIN zero-shot setting?```
>
> **A6**:
> We hypothesize the reason comes from two aspects.
> a) Even without video pre-training, *GR-1 w/o pretraining* utilized the video prediction to help the training of the action prediction.
> b) We used pre-trained MAE and CLIP, which have both been pretrained on large-scale datasets, to encode image and text, respectively.
> Therefore, *GR-1 w/o pretraining* naturally inherits the generalization capability from MAE and CLIP. However, we note that the performance of *GR-1 w/o pretraining* is not robust on real robots as shown in Appendix A.4.
>
> **Q7**: ```"283M parameters, in which 46M of them are trainable" -- to clarify, this means that the majority of the parameters are the frozen MAE and text encoder parameters?```
>
> **A7**:
> Yes, it is correct.
> MAE and CLIP, which have both been pre-trained on large-scale datasets, are powerful encoders.
> By using them for encoding images and texts respectively, GR-1 is able to achieve strong performance in both simulation and real robots.
>
> **Q8**: ```What is the control frequency of the real robot during the execution of the policy?```
>
> **A8**:
> The duration for model inference is about 270ms on our real robot platform which has a single NVIDIA 2060 GPU.
> As a reference, RT-1 runs at 3Hz [1].
> We use a positional controller to control the robot to the pose predicted by the model.
> The controller frequency is 100Hz.
>
> [1] Brohan, Anthony, et al. "RT-1: Robotics transformer for real-world control at scale." arXiv preprint arXiv:2212.06817 (2022).

---

### Author Response · Authors · 2023-11-23
**General Response**

We would like to thank all the reviewers for the constructive comments.
We work very hard to revise the paper accordingly.
Specifically:
- We added **comparison with R3M in both simulation and the real world** (Sec. 4.1 and 4.2).
- We added real robot experiments on **end-to-end object transportation and articulated object manipulation** (Sec. 4.2).
- We performed experiments on **generalization to unseen objects in the real world** (Sec. 4.2).
- We included experiments on **generalization to unseen language instructions** (Sec. 4.1).
- We fixed all the mentioned typos in the paper.

An updated version of the paper is uploaded and major changes are highlighted in blue.

---

### Meta-Review · Area_Chair_bp25 · 2023-12-10

**Metareview:**

The paper is concerned with understanding the utility of large autoregressive models trained on video and action data for the purpose of robot manipulation. In particular, the author propose to train contemporary GPT models on sequential data where they consume video frames, text, and states, and produce actions and future states. They use this model to perform language conditioned mobile manipulation both in simulation, CALVIN, as well as on a real robot, and demonstrate competitive performance in these settings.

All reviewers appreciate the use of large egocentric video datasets to obtain a representation that can be used for manipulation. Although the idea not being new, the authors appreciate the concrete system presented, and its simplicity, i.e. autoregressive formulation. Further, some of the reviewers appreciate results being presented both in simulation, CALVIN benchmark, as well as a real robot. Finally, the presented results are strong and superior to other contemporary work.

Some of the concerns raised by the reviewers are with regards to missing baselines, the simplicity of the real world experiments, and the generalization of the algorithm.

**Justification For Why Not Higher Score:**

The ratings are mixed 3, 5, 6, and 8. Due to some of the voiced reservations the ACs believe the paper isn't suitable for a spotlight.

**Justification For Why Not Lower Score:**

The paper has received mixed reviews: 1x reject, 1 x borderline reject, 1 x borderline accept, and 1 x accept.

The ACs feel that the paper presents a timely approach to an important problem. All reviewers appreciate the simplicity and power of the approach, the empirical insight of using egocentric video to learn representations for manipulation (this isn't a new lesson, but the formulation as an autoregressive model is new and promising). Further, the empirical evaluation is done both in simulation and real, which is considered extensive. In these two setups, the authors present analysis that tests against generalization w.r.t novel environments, novel language, etc. In all these experiments, the presented approach is strong, and in many cases by a margin. All this substantiates the approach.

The main criticism against the paper, as elaborated in the reject and borderline reject reviews has to do with experiments. First, two reviewers ask for more baselines. Although in AC eyes some of these requests are unsubstantiated, as the requested baselines fall within contemporaneous works, the authors compare against R3M in the rebuttal, asked for by two reviewers, and demonstrate superior performance. Given that the approach fares quite well against other approaches by a margin, the ACs feel the results are convincing enough.

Further, the reviewers have reservations re generalization results on CALVIN. But as explained in the rebuttal, the authors use the standard methodology of CALVIN, that tests against novel environments and, thanks to the addition to CALVIN by the authors, test against novel paraphrasings of instructions. The ACs believe that some of the reviews are due to misunderstandings, and find the clarification in the rebuttal satisfactory.

Finally, some of the reviewers find the real world experiments simplistic. The ACs believe that the response of the authors of additional experiments, object transportation, addresses some of these concerns.

---

### Decision · Program_Chairs · 2024-01-16

Accept (poster)